# PomX, a ParA/MinD ATPase activating protein, is a triple regulator of cell division in *Myxococcus xanthus*

Dominik Schumacher[1], Andrea Harms[1], Silke Bergeler[2], Erwin Frey[2], Lotte Søgaard-Andersen[1]*

[1]Department of Ecophysiology, Max Planck Institute for Terrestrial Microbiology, Karl-von-Frisch, Marburg, Germany; [2]Arnold Sommerfeld Center for Theoretical Physics and Center for NanoScience, Department of Physics, Ludwig-Maximilians-Universität München, München, Germany

**Abstract** Cell division site positioning is precisely regulated but the underlying mechanisms are incompletely understood. In the social bacterium *Myxococcus xanthus,* the ~15 MDa tripartite PomX/Y/Z complex associates with and translocates across the nucleoid in a PomZ ATPase-dependent manner to directly position and stimulate formation of the cytokinetic FtsZ-ring at midcell, and then undergoes fission during division. Here, we demonstrate that PomX consists of two functionally distinct domains and has three functions. The N-terminal domain stimulates ATPase activity of the ParA/MinD ATPase PomZ. The C-terminal domain interacts with PomY and forms polymers, which serve as a scaffold for PomX/Y/Z complex formation. Moreover, the PomX/PomZ interaction is important for fission of the PomX/Y/Z complex. These observations together with previous work support that the architecturally diverse ATPase activating proteins of ParA/MinD ATPases are highly modular and use the same mechanism to activate their cognate ATPase via a short positively charged N-terminal extension.

*For correspondence:
sogaard@mpi-marburg.mpg.de

**Competing interests:** The authors declare that no competing interests exist.

## Introduction

Accurate positioning of the cell division site ensures the formation of daughter cells of correct size and shape. In most bacteria, cell division initiates with positioning of the tubulin homolog FtsZ at the incipient division site (*Du and Lutkenhaus, 2019*). Subsequently, FtsZ polymerizes to form a dynamic ring-like structure, the so-called (Fts)Z-ring, which serves as a scaffold for recruitment of other components of the division machinery (*Du and Lutkenhaus, 2019*). Accordingly, systems that regulate division site positioning act at the level of Z-ring formation (*Eswara and Ramamurthi, 2017*). While the core components of the division machinery are conserved, the regulatory systems that ensure Z-ring positioning and, thus, the division site, are surprisingly diverse and also incompletely understood (*Eswara and Ramamurthi, 2017*). Interestingly, several of these systems have at their core a member of the ParA/MinD superfamily of P-loop ATPases that interacts with system-specific components to generate system-specific dynamic localization patterns that bring about correct Z-ring positioning (*Lutkenhaus, 2012*). These patterns include pole-to-pole oscillations in the *Escherichia coli* MinCDE system, bipolar gradient formation in the *Caulobacter crescentus* MipZ/ParB system, polar localization in the *Bacillus subtilis* MinCDJ/DivIVA system, and translocation across the nucleoid in the PomXYZ system of *Myxococcus xanthus* (*Treuner-Lange and Søgaard-Andersen, 2014*; *Schumacher and Søgaard-Andersen, 2017*).

ParA/MinD ATPases are key players in orchestrating the subcellular organization of bacterial cells and are not only involved in division site positioning but also in chromosome and plasmid segregation as well as positioning of other macromolecular complexes (*Lutkenhaus, 2012*). The function of

ParA/MinD ATPases critically depends on the ATPase cycle during which they alternate between a monomeric form and a dimeric ATP-bound form (*Leonard et al., 2005*; *Hu et al., 2003*; *Wu et al., 2011*; *Scholefield et al., 2011*). Generally, ParA/MinD ATPases have a low intrinsic ATPase activity that is stimulated by a single cognate ATPase Activating Protein (AAP) (*Lutkenhaus, 2012*). While ParA/MinD ATPases share overall sequence conservation, the AAPs are much less conserved. The ParABS systems for chromosome and plasmid segregation and the *E. coli* MinCDE system have provided fundamental insights into how ParA/MinD ATPase/AAP pairs interact to establish dynamic localization patterns: The AAP ParB stimulates ATPase activity of ATP-bound ParA dimers bound non-specifically to DNA (*Leonard et al., 2005*; *Bouet and Funnell, 1999*; *Hester and Lutkenhaus, 2007*). In chromosome segregation systems, ParB binds to multiple *parS* sites at the origin of replication, forming a large complex (*Lin and Grossman, 1998*; *Yamaichi et al., 2007*; *Mohl and Gober, 1997*) while ATP-bound ParA dimers bind non-specifically to the nucleoid (*Castaing et al., 2008*; *Hester and Lutkenhaus, 2007*; *Leonard et al., 2005*; *Scholefield et al., 2011*). Upon replication, one of the duplicated ParB-*parS* complexes interacts with nucleoid-bound ParA dimers, thereby stimulating ATP hydrolysis and causing the release of ParA monomers from the nucleoid (*Bouet and Funnell, 1999*; *Ptacin et al., 2010*; *Schofield et al., 2010*; *Vecchiarelli et al., 2013*). Released ParA monomers undergo nucleotide exchange and rebind to the nucleoid (*Bouet and Funnell, 1999*; *Ptacin et al., 2010*; *Schofield et al., 2010*; *Vecchiarelli et al., 2013*). Repeated interactions between the large ParB-*parS* complex and nucleoid-bound ParA result in translocation of the ParB-*parS* complex to the opposite cell half (*Vecchiarelli et al., 2014*; *Lim et al., 2014*; *Ptacin et al., 2010*; *Schofield et al., 2010*). In the MinCDE system, the AAP MinE, which is non-homologous to ParB, stimulates ATPase activity of membrane- and ATP-bound MinD dimers (*Hu and Lutkenhaus, 2001*; *Lackner et al., 2003*; *Park et al., 2011*; *Hu and Lutkenhaus, 2003*; *Szeto et al., 2002*). In vivo, dimeric ATP-bound MinD forms a complex with the inhibitor of Z-ring formation MinC at the membrane (*de Boer et al., 1991*; *Hu and Lutkenhaus, 1999*; *Hu and Lutkenhaus, 2003*; *Hu et al., 1999*). Upon stimulation of MinD ATPase activity by MinE, the MinD/C complex is released from the membrane (*Hu et al., 2002*; *Hu and Lutkenhaus, 2001*; *Lackner et al., 2003*; *Park et al., 2012*). Subsequently, MinD undergoes nucleotide exchange and rebinds to the membrane together with MinC. These interactions ultimately result in the coupled pole-to-pole oscillations of the MinC/D complex and MinE (*Hu and Lutkenhaus, 1999*; *Raskin and de Boer, 1999*).

In the rod-shaped *M. xanthus* cells, Z-ring formation and positioning at midcell between two segregated chromosomes are stimulated by the tripartite PomX/Y/Z complex (*Schumacher et al., 2017*; *Treuner-Lange et al., 2013*; *Harms et al., 2013*). PomZ is a ParA/MinD ATPase while PomX and PomY separately have AAP activity and function to synergistically stimulate the low intrinsic ATPase activity of DNA- and ATP-bound dimeric PomZ (*Schumacher et al., 2017*; *Treuner-Lange et al., 2013*). PomX and PomY are non-homologous, and share homology with neither ParB nor MinE. The Pom proteins form a dynamically localized complex with an estimated size of ~15 MDa in vivo that is visible as a cluster by epifluorescence microscopy (*Schumacher et al., 2017*). This complex associates with the nucleoid via PomZ, and early during the cell cycle, it is positioned on the nucleoid away from midcell, that is off-center, close to the new cell pole (*Schumacher et al., 2017*). Subsequently, the complex translocates by biased random motion on the nucleoid to the midnucleoid, which coincides with midcell. At midnucleoid, the PomXYZ complex undergoes constrained motion and stimulates Z-ring formation. Intriguingly, during cell division, the PomX/Y/Z complex undergoes fission, with the two 'portions' segregating to the two daughters (*Schumacher et al., 2017*).

The Pom proteins interact in all three pairwise combinations in vitro and PomX is essential for cluster formation by PomY and PomZ in vivo (*Schumacher et al., 2017*): In vitro PomX spontaneously polymerizes in a cofactor-independent manner to form filaments, and alone forms a cluster in vivo. PomY bundles PomX filaments in vitro; in vivo PomY is recruited by PomX to form a PomX/Y complex, which is not associated with the nucleoid and stalled somewhere in cells. ATP- and DNA-bound PomZ dimers, but not monomeric PomZ, are recruited to the PomX/Y complex by interactions with PomX as well as PomY, resulting in the association of the PomX/Y/Z complex to the nucleoid. Due to the AAP activity of PomX and PomY, PomZ is rapidly turned over in the PomX/Y/Z complex and released in a monomeric form to the cytosol. Motion of the PomX/Y/Z complex depends on non-specific DNA binding and ATP hydrolysis by PomZ (*Schumacher et al., 2017*). Translocation to midnucleoid and constrained motion at midnucleoid arise from the continuous turnover of PomZ in the complex together with the diffusive PomZ flux on the nucleoid into the PomX/

Y/Z complex (*Schumacher et al., 2017*). Finally, while all three Pom proteins are important for Z-ring formation at midcell, PomY and PomZ in the PomX/Y/Z complex are thought to be directly involved in recruiting FtsZ to the division site based on protein-protein interaction analyses (*Schumacher et al., 2017*).

The Pom system displays unusual spatiotemporal dynamics and is also unusual because it incorporates two AAPs. It remains elusive how PomX and PomY stimulate PomZ ATPase activity. With the long-term goal to understand the spatiotemporal dynamics of the Pom system, we focused on the function of PomX in vivo and in vitro. Here, we show that PomX is a triple regulator of cell division and displays three activities: The AAP activity resides in the N-terminal domain, the C-terminal domain serves as a scaffold for PomX/Y/Z complex formation, and the PomX/Z interaction is important for PomX/Y/Z complex fission during division. Moreover, our findings support the notion that that AAPs of ParA/MinD ATPases use the same mechanism to activate their cognate ATPase.

## Results

### PomX consists of two functional domains with distinct functions

PomX and PomY co-occur with PomZ in Cystobacterineae (*Schumacher et al., 2017*). PomZ sequences are highly conserved, whereas PomX and PomY sequences are more divergent (*Figure 1A*). In *M. xanthus* PomX, the region from residues 27 to 182 is Ala and Pro-rich (23% Pro; 19% Ala) and the region from residues 222 to 401 contains a coiled-coil domain (*Figure 1B*). Although PomX homologs are of different length, their overall architecture is similar and with a high level of similarity in the C-terminal regions while the N-terminal regions vary in length and similarity (*Figure 1B*). To analyze how PomX functions, we divided PomX into two parts. From hereon, we refer to these two parts as the N-terminal domain (PomX$^N$, residues 1–213) and the C-terminal domain (PomX$^C$, residues 214–404) (*Figure 1C*).

We fused the two PomX domains to mCherry (mCh) and expressed them ectopically (*Figure 1—figure supplement 1A*). mCh-PomX$^N$, mCh-PomX$^C$, and full-length mCh-PomX$^{WT}$ accumulated at or above PomX$^{WT}$ levels in Δ*pomX* and *pomX$^+$* cells (*Figure 1—figure supplement 1B*). WT *M. xanthus* cells have a cell length of 8.0 ± 1.8 μm (mean ± standard deviation (STDEV)), while Δ*pomX* cells are filamentous with a length of 13.1 ± 6.1 μm and also generate DNA-free minicells (*Figure 1D*). mCh-PomX$^{WT}$ complemented the division defect of the Δ*pomX* mutant in agreement with previous observations (*Schumacher et al., 2017*), while the truncated variants did not.

mCh-PomX$^{WT}$ formed a single well-defined cluster in 93% and 94% of Δ*pomX* and WT cells, respectively (*Figure 1E*). These clusters localized in the off-center position (defined as clusters outside the midcell region at 50 ± 5% of cell length) in short cells and at midcell in long cells (*Figure 1E*). The truncated mCh-PomX variants displayed diffuse localization in Δ*pomX* cells (*Figure 1E*). However, mCh-PomX$^C$ but not mCh-PomX$^N$ formed a single cluster in 90% of *pomX$^+$* cells and localized as mCh-PomX$^{WT}$ (*Figure 1E*). This cluster formation by mCh-PomX$^C$ was independent of PomY and PomZ (*Figure 1F* and *Figure 1—figure supplement 1B*), supporting that mCh-PomX$^C$ is integrated into the PomX/Y/Z cluster via interaction with PomX. As described (*Schumacher et al., 2017*), the PomX clusters were more elongated in the absence of PomY (*Figure 1F*). The incorporation of PomX$^C$ into the PomX/Y/Z complex interfered with neither PomX/Y/Z complex formation nor function (*Figure 1D,E*). We conclude that both PomX domains are essential for function and that mCh-PomX$^C$ can integrate into the PomX/Y/Z complex via interaction with PomX while mCh-PomX$^N$ cannot.

### PomX AAP activity resides in PomX$^N$

We used the bacterial adenylate cyclase two-hybrid system (BACTH) to test for protein-protein interactions involving the two PomX domains. In agreement with previous observations using purified proteins, full-length PomX$^{WT}$ self-interacted and interacted with PomY (*Figure 2A*). PomX$^C$ self-interacted and also interacted with full-length PomX and PomY, while PomX$^N$ neither self-interacted nor interacted with PomX$^{WT}$, PomX$^C$, or PomY. To test for interactions with PomZ, we used the two PomZ variants PomZ$^{WT}$ and PomZ$^{D90A}$, the latter of which is locked in the DNA-binding, dimeric, ATP-bound form that interacts strongly with the PomX/PomY cluster in vivo (*Schumacher et al.,*

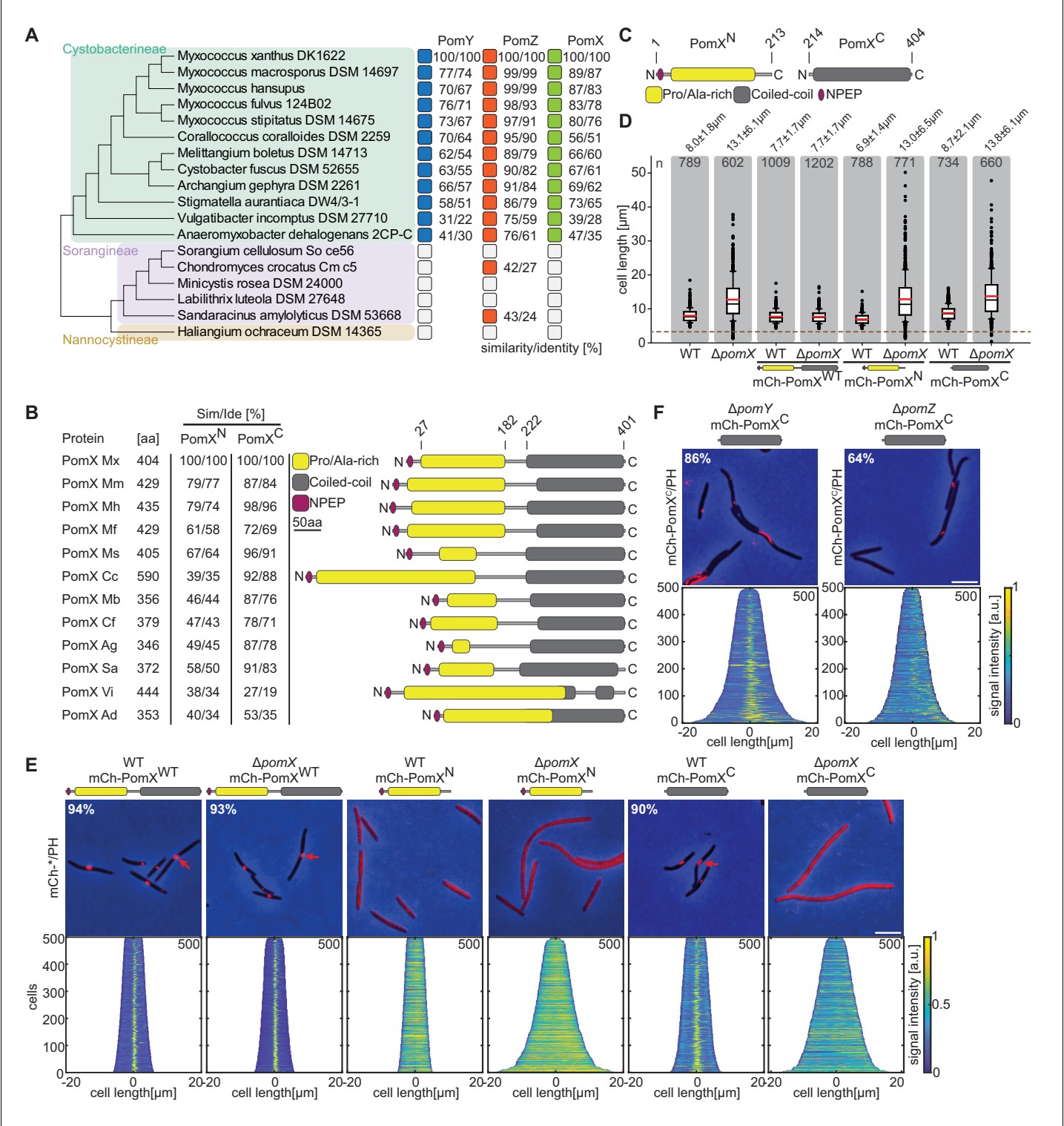

**Figure 1.** PomX consists of two domains that are both required for function. (**A**) Similarity and identity analysis of PomX, PomY, and PomZ homologs. The three Myxococcales suborders are indicated. An open box indicates that a homolog is not present. (**B**) Similarity and identity of PomX domains in different PomX homologs. Similarity and identity were calculated based on the domains of *M. xanthus* PomX shown in C. (**C**) PomX truncations used in this study. Numbers on top indicate the start and stop positions of the truncations relative to full-length PomX^WT. (**D**) Cell length distribution of cells of indicated genotypes. Cells below stippled line are minicells. Numbers indicate mean cell length±STDEV. In the boxplots, boxes include the 25th and the 75th percentile, whiskers data points between the 10% and 90% percentile, outliers are shown as black dots. Black and red lines indicate the median and mean, respectively. Number of analyzed cells is indicated. In the complementation strains, *pomX* alleles were expressed from plasmids

*Figure 1 continued on next page*

*Figure 1 continued*

integrated in a single copy at the *attB* site. (E) Fluorescence microscopy of cells of indicated genotypes. Phase-contrast and fluorescence images of representative cells were overlayed. Numbers indicate fraction of cells with fluorescent clusters. Demographs show fluorescence signals of analyzed cells sorted according to length and with off-center signals to the right. Numbers in upper right indicate number of cells used to create demographs. Scale bar, 5 µm. (F) Fluorescence microscopy of cells of indicated genotypes. Images of representative cells and demographs were created as in (E). Scale bar, 5 µm. For experiments in D, E and F similar results were obtained in two independent experiments.

The online version of this article includes the following source data and figure supplement(s) for figure 1:

**Source data 1.** Source Data for *Figure 1D*.
**Figure supplement 1.** PomX variants accumulate in *M.xanthus*.
**Figure supplement 1—source data 1.** Source data for *Figure 1—figure supplement 1B*.

---

*2017*). PomX$^{WT}$ and PomX$^N$ interacted with PomZ$^{WT}$ and PomZ$^{D90A}$ while PomX$^C$ did not; also, PomZ$^{D90A}$ interacted more strongly with PomX$^N$ than PomZ$^{WT}$ (*Figure 2A*).

To test for interactions between the Pom proteins in vitro, we purified tagged variants of PomX$^{WT}$, PomX$^N$, PomX$^C$, PomY, and PomZ (*Figure 2—figure supplement 1A*). We confirmed by negative stain transmission electron microscopy (TEM) that PomX$^{WT}$-His$_6$ formed filaments that were bundled by PomY-His$_6$, while PomY-His$_6$ on its own did not form higher order structures (*Figure 2B*; *Schumacher et al., 2017*). Consistently, when analyzed separately in high-speed centrifugation experiments, 93–95% of PomX$^{WT}$-His$_6$ and 29–35% of PomY-His$_6$ were recovered in the pellet fraction, whereas 71–86% and 48–49%, respectively of PomX$^{WT}$-His$_6$ and PomY-His$_6$ were in the pellet fraction when mixed in equimolar amounts (*Figure 2—figure supplement 1B,C,D*).

As noted for PomX$^{WT}$, PomX$^N$-His$_6$ (molecular weight (MW) of monomer: 24.3 kDa) migrated aberrantly in SDS-PAGE (*Figure 2—figure supplement 1A*). By size exclusion chromatography (SEC), the majority of PomX$^N$-His$_6$ eluted corresponding to a globular protein with a MW of ~136 kDa and in a smaller peak corresponding to a MW of ~306 kDa (*Figure 2—figure supplement 1E*). PomX$^N$-His$_6$ neither formed higher-order structures by TEM nor in high-speed centrifugation experiments (*Figure 2B* and *Figure 2—figure supplement 1C,D*). Because PomX$^N$ does not self-interact in BACTH, migrates aberrantly by SDS-PAGE, and is rich in Ala/Pro residues and, therefore, may not have a globular conformation, it is unclear whether SEC reflects the formation of PomX$^N$ oligomers. PomX$^C$-His$_6$ in SDS-PAGE migrated at the expected size (MW of monomer: 22.8 kDa) (*Figure 2—figure supplement 1A*); however, the protein could not be analyzed by SEC because it did not enter the matrix. Accordingly, PomX$^C$-His$_6$ spontaneously formed filaments visible by TEM and mostly accumulated in the pellet fraction in high-speed centrifugation experiments (*Figure 2B* and *Figure 2—figure supplement 1B,D*). PomY-His$_6$ bundled the PomX$^C$-His$_6$ filaments and was enriched in the pellet fraction in centrifugation assays in the presence of PomX$^C$-His$_6$ (*Figure 2B* and *Figure 2—figure supplement 1B*). Interactions between PomX$^N$-His$_6$ and PomX$^{WT}$-His$_6$, PomX$^C$-His$_6$ and PomY-His$_6$ were observed by neither TEM nor high-speed centrifugation (*Figure 2B* and *Figure 2—figure supplement 1C,D*). Finally, we confirmed in pull-down experiments using truncated PomX-Strep variants (*Figure 2—figure supplement 1A*) that PomX$^{WT}$-His$_6$ and PomY-His$_6$ interact with PomX$^C$-Strep (*Figure 2C*) but not with PomX$^N$-Strep (*Figure 2D*) and that PomX$^C$-His$_6$ did not interact with PomX$^N$-Strep (*Figure 2—figure supplement 1F*).

To test in vitro for interactions between PomX variants and PomZ, we used PomZ ATPase activity as a readout. First, we established a base-line for these analyses. The amount of hydrolyzed ATP increased with His$_6$-PomZ concentration in the absence of DNA (specific activity: 7 ± 1 ATP hr$^{-1}$, 4 µM His$_6$-PomZ) (*Figure 2E*). In the presence of increasing concentrations of non-specific herring sperm DNA, His$_6$-PomZ ATP hydrolysis was stimulated, reaching saturation at ~40 µg/ml DNA. At this concentration, His$_6$-PomZ ATP hydrolysis was stimulated twofold (specific activity: ~15 ATP hr$^{-1}$, 4 µM His$_6$-PomZ) (*Figure 2F*). In the presence of saturating concentrations of DNA (60 µg/ml), His$_6$-PomZ ATPase activity also increased with concentration (*Figure 2E*). For comparison, in an average *M. xanthus* cell with one chromosome, the DNA concentration is ~3400 µg/ml suggesting that PomZ in vivo works under fully DNA-saturating conditions.

PomX$^{WT}$-His$_6$ only stimulated His$_6$-PomZ ATPase activity in the presence of DNA (*Figure 2G*). Stimulation increased with increasing PomX$^{WT}$-His$_6$ concentrations (specific activity: 44 ± 7 ATP hr$^{-1}$ at 15 µM PomX$^{WT}$-His$_6$, 4 µM His$_6$-PomZ), and His$_6$-PomZ ATPase activity did not reach a plateau even at the highest PomX$^{WT}$-His$_6$ concentration (*Figure 2G*). Importantly, and in agreement with the

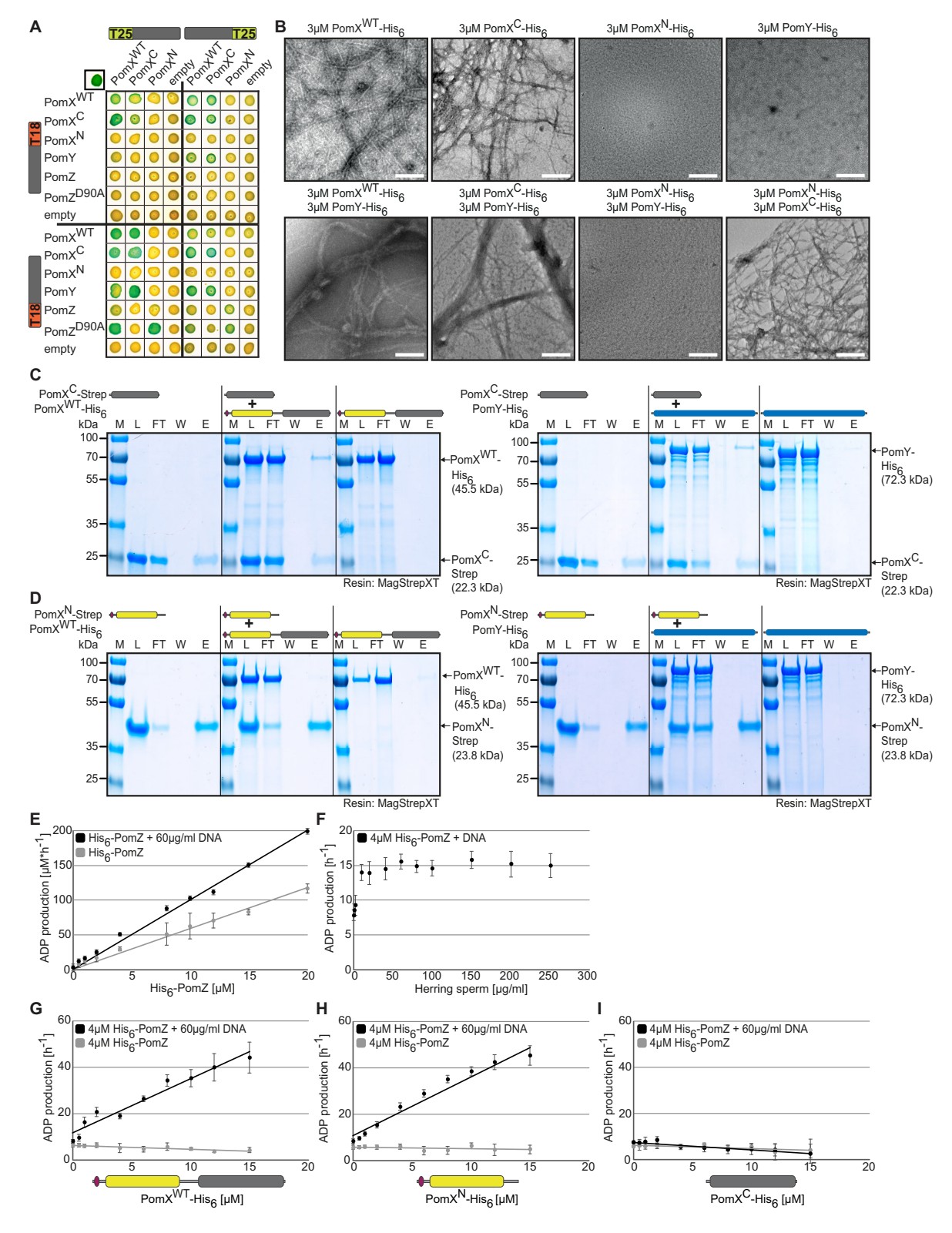

**Figure 2.** PomX[C] interacts with PomX and PomY while PomX[N] stimulates PomZ ATPase activity. (**A**) BACTH analysis of interactions between Pom proteins. The indicated protein fragments were fused to T18 and T25 as indicated. Blue colony indicates an interaction, white no interaction. Positive control in upper left corner, leucine zipper of GCN4 fused to T25 and T18. For negative controls, co-transformations with empty plasmids were performed. Images show representative results and were performed in three independent experiments. (**B**) TEM images of negatively stained purified

*Figure 2 continued on next page*

*Figure 2 continued*

proteins. Proteins were applied to the EM grids alone or after mixing in a 1:1 molar ratio as indicated before staining. Scale bar, 200 nm. Images show representative results of several independent experiments. (C, D) In vitro pull-down experiments with purified PomX$^C$-Strep, PomX$^N$-Strep, PomX$^{WT}$-His$_6$, and PomY-His$_6$. Instant Blue-stained SDS-PAGE shows load (L), flow-through (FL), wash (W), and elution (E) fractions using MagStrep XT beads in pull-down experiments with 10 µM of indicated proteins alone or pre-mixed as indicated on top. Molecular size markers are shown on the left and proteins analyzed on the right together with their calculated MW. Note that PomX$^{WT}$-His$_6$ (*Schumacher et al., 2017*) and PomX$^N$-Strep migrate aberrantly and according to a higher MW. All samples in a panel were analyzed on the same gel and black lines are included for clarity. Experiments were repeated in two independent experiments with similar results. (E–I) His$_6$-PomZ ATPase activity. ADP production rate was determined in an NADH-coupled photometric microplate assay in the presence of 1 mM ATP at 32˚C. DNA and PomX variants were added as indicated. Spontaneous ATP hydrolysis and NADH consumption was accounted for by subtracting the measurements in the absence of His$_6$-PomZ. Data points show the mean±STDEV calculated from six independent measurements.

The online version of this article includes the following source data and figure supplement(s) for figure 2:

**Source data 1.** Source data for *Figure 2A*.
**Source data 2.** Source data for *Figure 2B*.
**Source data 3.** Source data for *Figure 2C*.
**Source data 4.** Source data for *Figure 2D*.
**Source data 5.** Source data for *Figure 2E*.
**Source data 6.** Source data for *Figure 2F*.
**Source data 7.** Source data for *Figure 2G*.
**Source data 8.** Source data for *Figure 2H*.
**Source data 9.** Source data for *Figure 2I*.
**Figure supplement 1.** Purification and analysis of Pom proteins.
**Figure supplement 1—source data 1.** Source data for *Figure 2—figure supplement 1A*.
**Figure supplement 1—source data 2.** Source data for *Figure 2—figure supplement 1B*.
**Figure supplement 1—source data 3.** Source data for *Figure 2—figure supplement 1C*.
**Figure supplement 1—source data 4.** Source data for *Figure 2—figure supplement 1D*.
**Figure supplement 1—source data 5.** Source data for *Figure 2—figure supplement 1E*.
**Figure supplement 1—source data 6.** Source data for *Figure 2—figure supplement 1F*.

BACTH analysis, PomX$^N$-His$_6$ stimulated His$_6$-PomZ ATP hydrolysis in the presence of DNA as efficiently as PomX$^{WT}$-His$_6$ (specific activity: 43 ± 4 ATP hr$^{-1}$ at 15 µM PomX$^N$-His$_6$, 4 µM His$_6$-PomZ) (*Figure 2H*). By contrast, PomX$^C$-His$_6$ did not stimulate His$_6$-PomZ ATPase activity (*Figure 2I*). The observations that PomX$^N$ does not spontaneously form filaments in vitro while full-length PomX does and these two PomX variants stimulate PomZ ATPase activity with equal efficiency provide evidence that PomX filament formation is not essential for AAP activity. These observations also strongly support that the PomX$^N$ domains in a PomX filament act independently of each other to stimulate PomZ ATPase activity.

Altogether, we conclude that (1) PomX consists of two domains with distinct functions that are both required for PomX activity in vivo; (2) PomX$^N$ interacts with PomZ and harbors the entire AAP activity; (3) PomX$^C$ is required and sufficient to mediate PomX self-interaction with spontaneous filament formation in vitro and PomX-dependent cluster incorporation in vivo; (4) PomX$^C$ interacts with PomY; and, (5) the PomX$^N$ domains in a PomX filament function independently of each other to stimulate PomZ ATPase activity.

## Two positively charged residues in PomX$^{NPEP}$ are important for division site and PomX/Y/Z cluster positioning at midcell

To define the PomX$^N$ region involved in AAP activity, we performed a detailed sequence analysis of the N-terminal domain of PomX homologs. This analysis revealed a stretch of highly conserved amino acids at the N-terminus (residues 1–22, from hereon PomX$^{NPEP}$) that is enriched in charged amino acids, six of which are positively charged (*Figures 1B* and *3A* and *Figure 3—figure supplement 1*).

To probe the role of PomX$^{NPEP}$, we generated PomX variants lacking residues 2–21 (PomX$^{\Delta2-21}$) with and without mCh; however, none of these variants accumulated in *M. xanthus*. Therefore, we performed Ala scanning of PomX$^{NPEP}$ in which all charged or hydrophilic residues were replaced by Ala. Then the mutant alleles replaced the *pomX$^{WT}$* allele at the *pomX* locus. All 11 PomX variants except for PomX$^{K3A}$ accumulated similarly to or at slightly lower or higher levels than PomX$^{WT}$

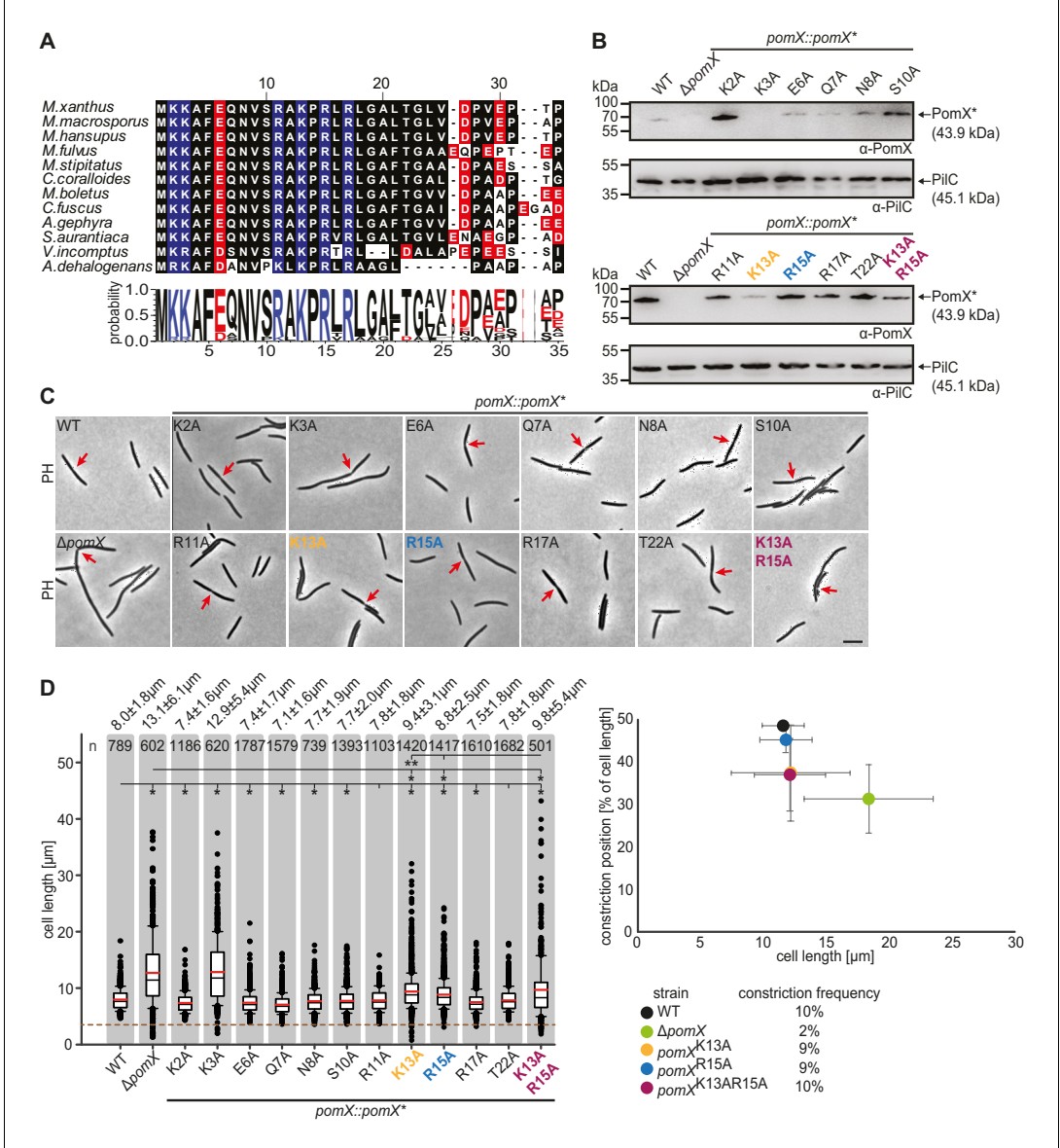

**Figure 3.** PomX^N harbors a conserved N-terminal peptide crucial for cell division site positioning at midcell. (A) Multiple sequence alignment of the conserved PomX N-terminus. Black background indicates similar amino acids. Positively and negatively charged residues are indicated in blue and red, respectively. Weblogo consensus sequence is shown below. (B) Western blot analysis of accumulation of PomX variants. Protein from the same number of cells was loaded per lane. Molecular mass markers are indicated on the left. PilC was used as a loading control. (C) Phase-contrast microscopy of strains of indicated genotypes. Representative cells are shown. Red arrows indicate cell division constrictions. Scale bar, 5 μm. (D) Analysis of cell length distribution and cell division constrictions of cells of indicated genotypes. Left panel, boxplot is as in *Figure 1D*. Number of cells analyzed is indicated at the top. *p<0.001; **p<0.05 in Mann-Whitney test. Right panel, cell division position in % of cell length is plotted as a function of cell length. Dots represent mean ± STDEV. Numbers below indicate cell division constriction frequency. In B, C, and D, similar results were obtained in two independent experiments.

The online version of this article includes the following source data and figure supplement(s) for figure 3:

**Source data 1.** Source data for *Figure 3B*.

**Source data 2.** Source data for *Figure 3D*.

**Figure supplement 1.** PomX homologs are highly conserved.

(*Figure 3B*). Most strains had a cell length similar to WT and cell division constrictions at midcell (*Figure 3C,D*). As expected, *pomX*^K3A cells were similar to Δ*pomX* cells (but see also details below about mCh-PomX^K3A). More importantly, the *pomX*^K13A and *pomX*^R15A mutants generated

filamentous cells and minicells, but the filamentous cells were shorter than $\Delta pomX$ cells (*Figure 3D* left panel). $pomX^{K13A}$ and $pomX^{R15A}$ cells had a cell division constriction frequency similar to WT (*Figure 3D* right panel), but these constrictions were mostly not at midcell (*Figure 3C,D* right panel). The $pomX^{K13AR15A}$ double mutant had a more pronounced filamentation phenotype than the single mutants, formed minicells, had a constriction frequency similar to WT, and mostly with the constrictions away from midcell (*Figure 3C,D*). PomX$^{K13AR15A}$ accumulated similarly to PomX$^{WT}$ (*Figure 3B*). We conclude that Lys13 and Arg15 in PomX$^{NPEP}$ are important for PomX function. Because the PomX$^{K13A}$, PomX$^{R15A}$, and PomX$^{K13AR15A}$ variants caused defects distinct from the $\Delta pomX$ mutant, i. e., cells are shorter, and with more constrictions, we conclude that substitution of Lys13 and/or Arg15 does not result in a complete loss of PomX function (see Discussion).

Next, we determined the localization of the PomX$^{NPEP}$ variants with Ala substitutions using mCh fusion proteins ectopically expressed in the $\Delta pomX$ mutant. All 11 fusion proteins with a single substitution as well as mCh-PomX$^{K13AR15A}$ accumulated at the same level as mCh-PomX$^{WT}$ in *M. xanthus* (*Figure 4—figure supplement 1A*). Most strains including the one expressing mCh-PomX$^{K3A}$ (cell length: $7.9 \pm 1.8$ μm) had a cell length similar to WT (cell length: $7.7 \pm 1.7$ μm) and cell division constrictions at midcell (*Figure 4—figure supplement 1B*) demonstrating that Lys3 is not important for PomX function. More importantly, mCh-PomX$^{K13A}$, mCh-PomX$^{R15A}$, and mCh-PomX$^{K13AR15A}$ formed clusters in vivo; however, these were mostly not at midcell and in the case of mCh-PomX$^{K13AR15A}$ ~50% localized in the DNA-free subpolar regions while this localization pattern was not observed for mCh-PomX$^{WT}$ (*Figure 4A* and *Figure 4—figure supplement 1C*). The clusters formed by these three mCh-PomX variants, similarly to those of mCh-PomX$^{WT}$, colocalized with cell division constrictions. Overall, these observations demonstrate that mCh-PomX$^{K13A}$, mCh-PomX$^{R15A}$, and mCh-PomX$^{K13AR15A}$ are functional in forming clusters, defining the division site, and stimulating division but cannot correctly position the division site at midcell. All other PomX variants with substitutions in PomX$^{NPEP}$, including mCh-PomX$^{K3A}$, localized as mCh-PomX$^{WT}$ (*Figure 4—figure supplement 1B*).

Because recruitment of the PomX/Y cluster to the nucleoid and PomX/Y/Z cluster localization at midcell depend on PomZ, these observations pointed in the direction that the PomZ and PomX interaction involves Lys13 and Arg15. To test for an interaction between PomZ and PomX$^{NPEP}$ variants in vivo, we explored PomX$^{K13AR15A}$ in more detail and took advantage of the ATP-locked PomZ$^{D90A}$ variant. In the presence of PomX$^{WT}$, PomX/Y/Z$^{D90A}$ clusters are randomly positioned on the nucleoid and rarely at midcell due to lack of PomZ ATPase activity (*Schumacher et al., 2017*; *Figure 4B*). We confirmed that in the absence of PomX, PomZ$^{D90A}$-mCh did not form clusters and instead colocalized with the nucleoid, while PomZ$^{D90A}$-mCh still formed clusters in the absence of PomY (*Figure 4B* and *Figure 4—figure supplement 2A*). In the presence of PomX$^{K13AR15A}$, PomZ$^{D90A}$-mCh formed clusters (*Figure 4B*). Because PomZ also interacts with PomY, we speculated that this cluster incorporation resulted from PomY recruiting PomZ$^{D90A}$-mCh. Indeed, upon additional deletion of *pomY*, PomZ$^{D90A}$-mCh no longer formed clusters and colocalized with the nucleoid when PomX$^{K13AR15A}$ was the only source of PomX (*Figure 4B*).

An active PomY-mCh fusion did not form clusters in the absence of PomX but formed clusters in the presence of PomX$^{K13AR15A}$ (*Figure 4C* and *Figure 4—figure supplement 2B*). As expected, in the presence of PomX$^{K13AR15A}$, the PomY-mCh clusters generally localized away from midcell as opposed to clusters in the presence of PomX$^{WT}$.

Altogether, these observations support that PomX$^{NPEP}$ is required for the interaction between PomX and PomZ but for neither PomX self-interaction nor PomX interaction with PomY. Moreover, they support that the PomX/Y/Z clusters formed in the presence of PomX$^{K13AR15A}$ are proficient in stimulating cell division but deficient in efficiently localizing to midcell, consistent with the PomX/PomZ interaction being perturbed. Finally, they support that PomZ in its ATP-bound dimeric form can be recruited independently by PomX and PomY to the PomX/Y complex.

## PomX$^{NPEP}$ is required and sufficient for stimulation of PomZ ATP hydrolysis

We generated PomX constructs for BACTH analysis that were either truncated for PomX$^{NPEP}$ (PomX$^{\Delta 2-21}$ and PomX$^{N-\Delta 2-21}$) or contained substitutions in PomX$^{NPEP}$ (PomX$^{K13AR15A}$ and PomX$^{N-K13AR15A}$). PomX$^{K13AR15A}$ and PomX$^{\Delta 2-21}$ self-interacted, and also interacted with PomX$^{WT}$ and PomY (*Figure 5A*). Consistently, in vitro PomX$^{K13AR15A}$-His$_6$ formed filaments that were bundled by PomY-His$_6$, accumulated in the pellet fraction after high-speed centrifugation, and brought PomY-His$_6$ to

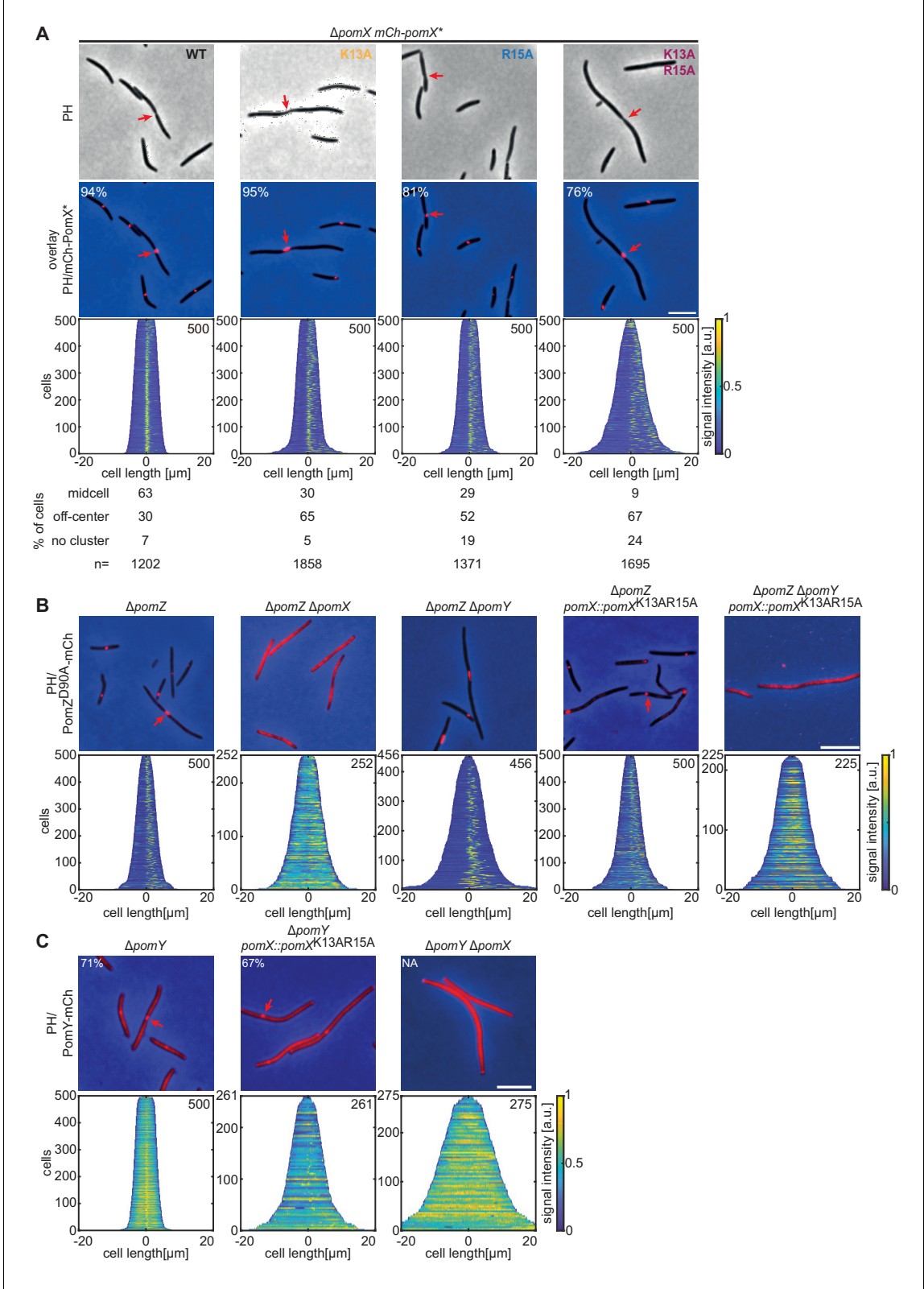

**Figure 4.** PomX$^{K13AR15A}$ forms clusters and interacts with PomY but not with PomZ in vivo. (A-C) Fluorescence microscopy of cells of indicated genotypes. Phase-contrast (PH) images and/or overlays of fluorescence images and PH of representative cells. Red arrows indicate division constrictions. Scale bar, 5 μm. In **A**, numbers in overlays indicate fraction of cells with a cluster and numbers below indicate localization patterns in % and number of cells analyzed. Demographs are as in *Figure 1E*. Similar results were observed in two independent experiments.

*Figure 4 continued on next page*

Figure 4 continued

The online version of this article includes the following source data and figure supplement(s) for figure 4:

**Source data 1.** Source data for *Figure 4A*.
**Source data 2.** Source data for *Figure 4B*.
**Source data 3.** Source data for *Figure 4C*.
**Figure supplement 1.** The PomX$^{K13AR15A}$ variant is impaired in function.
**Figure supplement 1—source data 1.** Source data for *Figure 4—figure supplement 1A*.
**Figure supplement 1—source data 2.** Source data for *Figure 4—figure supplement 1C*.
**Figure supplement 2.** Western blot analysis of PomY-mCh and PomZ$^{D90A}$-mCh accumulation.
**Figure supplement 2—source data 1.** Source data for *Figure 4—figure supplement 2A*.
**Figure supplement 2—source data 2.** Source data for *Figure 4—figure supplement 2B*.

the pellet fraction similarly to PomX$^{WT}$-His$_6$ (*Figure 5B,C*). As expected, in the BACTH neither PomX$^{N-\Delta2-21}$ nor PomX$^{N\_K13AR15A}$ interacted with PomX$^{WT}$ and PomY (*Figure 5A*). Importantly, all four PomX$^{NPEP}$ variants (PomX$^{\Delta2-21}$, PomX$^{N\_\Delta2-21}$, PomX$^{K13AR15A}$ and PomX$^{N\_K13AR15A}$) were dramatically reduced in interaction with PomZ and PomZ$^{D90A}$ (*Figure 5A*; see also *Figure 2A*). Altogether, these findings further support that PomX$^{NPEP}$ is specifically important for the PomX/PomZ interaction and not for PomX/PomX and PomX/PomY interactions.

Next, we tested whether PomX$^{NPEP}$ is important for PomX AAP activity. Because His$_6$-tagged truncated PomX$^{NPEP}$ variants could not be overexpressed in *E. coli*, we focused on the PomX$^{K13AR15A}$-His$_6$ and PomX$^{N\_K13RAR15A}$-His$_6$ variants (*Figure 2—figure supplement 1A*). PomX$^{N\_K13AR15A}$-His$_6$ (calculated MW: 24.1 kDa) behaved similarly to PomX$^{N}$-His$_6$ in SEC and eluted as a single peak corresponding to a globular protein of ~136 kDa (*Figure 2—figure supplement 1E*). Remarkably, neither PomX$^{K13AR15A}$-His$_6$ nor PomX$^{N\_K13AR15A}$-His$_6$ stimulated PomZ ATPase activity (*Figure 5D*). Consequently, we tested whether a peptide consisting of the 22 PomX$^{NPEP}$ residues alone is sufficient to stimulate His$_6$-PomZ ATPase activity. The PomX$^{NPEP}$ peptide alone stimulated His$_6$-PomZ ATPase activity in the presence of DNA (specific activity: 27 ± 5 ATP hr$^{-1}$ at 15 μM PomX$^{NPEP}$, 4 μM His$_6$-PomZ) while a PomX$^{NPEP}$ peptide with the K13AR15A substitutions did not (*Figure 5E*). We conclude that PomX$^{NPEP}$ is required and sufficient for stimulation of PomZ ATPase activity by PomX.

## The PomX/PomZ interaction is important for PomX/Y/Z cluster fission during division

Twenty-four percent of cells containing mCh-PomX$^{K13AR15A}$ lacked a visible cluster compared to only 6% in the presence of mCh-PomX$^{WT}$ (*Figure 4A*). Tagged and untagged PomX$^{K13AR15A}$ accumulate at the same level as tagged and untagged PomX$^{WT}$ (*Figure 3B* and *Figure 4—figure supplement 1A*), suggesting that this difference in cluster formation is not caused by differences in gene expression or protein stability. We, therefore, investigated whether mCh-PomX$^{K13AR15A}$ causes a cluster fission defect during cell division.

In the presence of mCh-PomX$^{WT}$, ~80% of divisions are accompanied by symmetric or asymmetric cluster fission, with each portion of a divided cluster segregating to a daughter (*Figure 6A,B*). In the remaining ~20%, cluster splitting did not visibly occur, the undivided cluster segregated to one of the daughters, and 'empty' daughter cells eventually regenerated a cluster that was visible after ~2 hr (*Figure 6A*). mCh-PomX$^{K13AR15A}$ clusters showed the same three patterns during division. However, cluster fission and segregation to daughters occurred in only ~20% of cells (*Figure 6A,B*).

Because PomX$^{K13AR15A}$ has reduced PomZ AAP activity, we tested cluster fission in Δ*pomZ* cells and in cells containing PomZ$^{D90A}$. In Δ*pomZ* cells, division occurs at a reduced frequency but still over the PomX/Y cluster (*Schumacher et al., 2017*). In these cells, mCh-PomX$^{WT}$ clusters rarely underwent fission and mostly segregated into one daughter (*Figure 6A,B*). In cells with PomZ$^{D90A}$, divisions also occurred over the cluster, and ~80% of clusters underwent fission during division and segregated to both daughters (*Figure 6A,B*). Altogether, these observations suggest that the interaction between PomZ and PomX is important for PomX/Y/Z cluster fission, while ATP hydrolysis by PomZ is not.

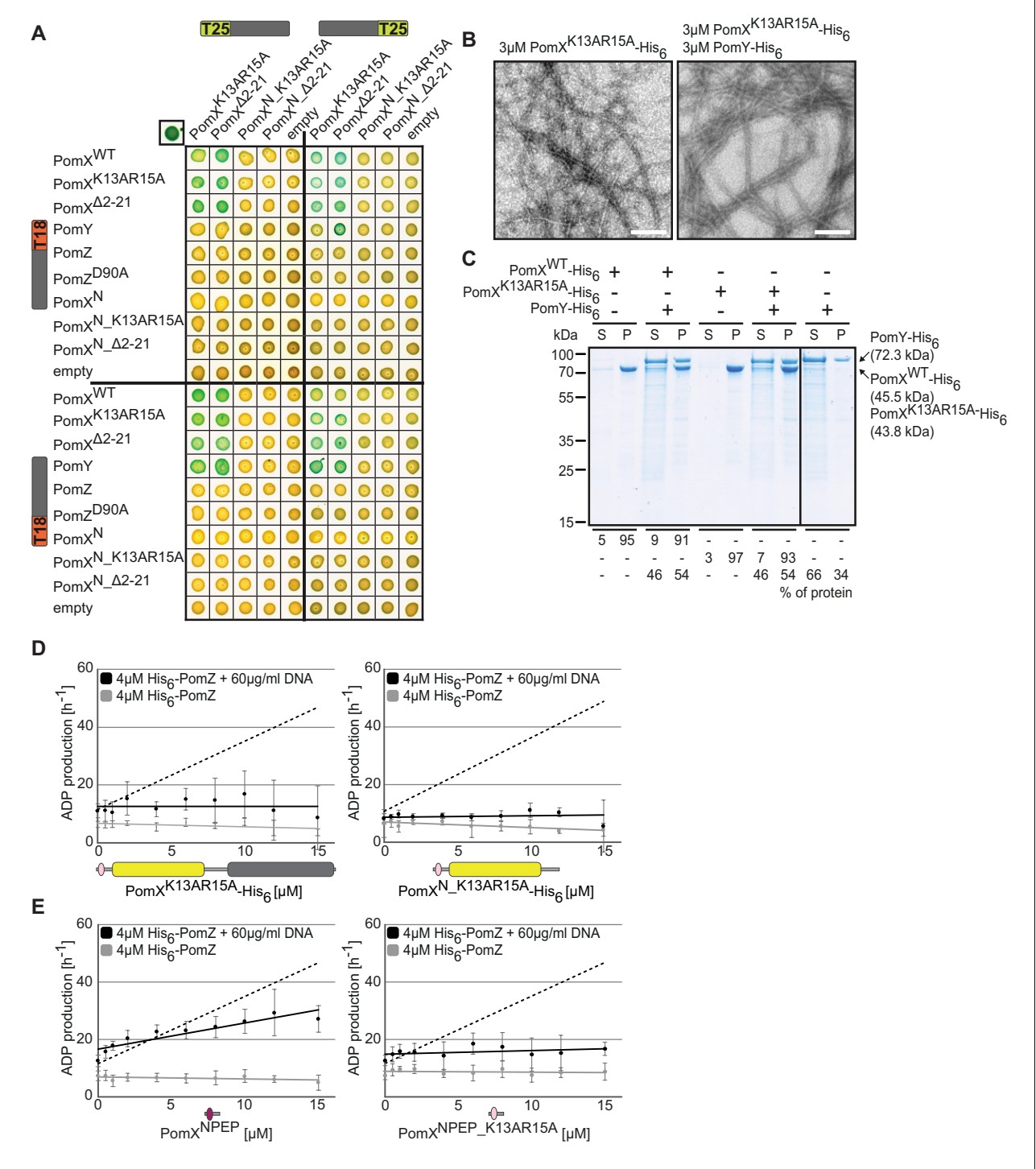

**Figure 5.** PomX AAP activity resides in PomX^NPEP. (**A**) BACTH analysis of interactions between Pom proteins and PomX variants. Experiments were performed in parallel with those in *Figure 2A*. For presentation purposes, the results for PomX^WT and PomX^N T25 fusion proteins and their interaction with PomZ and PomZ^D90A T18 fusion proteins were not included but are included in *Figure 2A*. Images show representative results and similar results were obtained in three independent experiments. (**B**) TEM images of negatively stained purified proteins. Experiments were done as in *Figure 2B*. Scale bar, 200 nm. Images show representative results of several independent experiments. (**C**) Sedimentation assays with indicated purified proteins. Proteins were analyzed at a concentration of 3 µM alone or in combination. After high-speed centrifugation, proteins in the supernatant (S) and pellet (P) fractions were separated by SDS-PAGE and stained with Instant Blue. Molecular size markers are shown on the left and analyzed proteins on the right including their calculated MW. Numbers below indicate % of proteins in different fractions. Similar results were obtained in two independent experiments. All samples were analyzed on the same gel; the black line indicates that lanes were removed for presentation purposes. (**D, E**) His_6-PomZ

*Figure 5 continued on next page*

*Figure 5 continued*

ATPase activity. Experiments were done and analyzed as in *Figure 2E–I* in the presence or absence of DNA and the indicated proteins and peptides. Data points show the mean±STDEV calculated from six independent measurements. In (D), stippled lines indicate the regression of the ADP production rate in the presence of PomX$^{WT}$-His$_6$ (left, *Figure 2G*) and PomX$^N$-His$_6$ (right, *Figure 2H*). In E, the stippled line indicates the regression of the ADP production rate in the presence of PomX$^{WT}$-His$_6$.

The online version of this article includes the following source data for figure 5:

**Source data 1.** Source data for *Figure 5A*.
**Source data 2.** Source data for *Figure 5B*.
**Source data 3.** Source data for *Figure 5C*.
**Source data 4.** Source data for *Figure 5D*.
**Source data 5.** Source data for *Figure 5E*.

## Discussion

In the present study, we used in vivo and in vitro approaches to functionally dissect the cell division regulatory protein PomX. We demonstrate that PomX is composed of two domains and has three functions in vivo. The N-terminal PomX$^N$ domain contains the PomZ AAP activity; the C-terminal PomX$^C$ domain is essential for PomX self-interaction, for the interaction with PomY, and functions as a scaffold for PomX/Y/Z cluster formation; and, the PomX/Z interaction, but not PomZ ATPase activity, is important for PomX/Y/Z cluster fission during division.

PomX$^N$, which is Ala/Pro-rich and, therefore, likely unstructured, interacts with PomZ and activates ATPase activity of DNA-bound PomZ in vitro as efficiently as PomX$^{WT}$. Moreover, a peptide comprising the N-terminal 22 residues of PomX (PomX$^{NPEP}$) is sufficient to activate PomZ ATPase activity. In PomX$^{NPEP}$, two positively charged residues (Lys13 and Arg15) are essential for AAP activity. Of note, the AAP activity of PomX$^{NPEP}$ was lower than that of PomX$^N$. Similarly, in the case of the ParB homologs Spo0J of *Thermus thermophilus* and SopB of plasmid F of *E. coli*, as well as MinE of *Neisseria gonorrhoeae*, peptides comprising the N-terminal 20, 52, and 22 residues, respectively are sufficient for AAP activity (*Ah-Seng et al., 2009*; *Ghasriani et al., 2010*; *Leonard et al., 2005*). In the case of the shorter Spo0J and MinE peptides, the AAP activity was also lower than for the full-length proteins (*Ghasriani et al., 2010*; *Leonard et al., 2005*), while the longer SopB peptide was as efficient as full-length SopB (*Ah-Seng et al., 2009*). Because BACTH analyses did not reveal an interaction between PomZ and PomX variants lacking PomX$^{NPEP}$ or containing the K13AR15A substitutions, we speculate that the lower AAP activity of PomX$^{NPEP}$ indicates that the context of PomX$^{N-PEP}$ might be important for its AAP activity; however, we cannot rule out that interactions between PomZ and PomX$^N$ beyond N$^{PEP}$ may also be important for PomZ ATPase activation.

PomX$^C$ with its predicted coiled-coil domain self-interacts and also interacts with PomX$^{WT}$ and PomY. Specifically, PomX$^C$, similarly to PomX$^{WT}$, formed filaments in vitro that were bundled by PomY. In vivo mCh-tagged PomX$^C$ integrated into clusters containing PomX$^{WT}$ but alone was not sufficient to form a cluster. By contrast, PomX$^N$ did not form or integrate into a cluster under any conditions tested; moreover, PomX$^N$ interacted with neither PomX$^{WT}$ nor PomY in BACTH or in vitro. Because PomX$^{WT}$ alone can form a cluster and is essential for PomX/Y/Z cluster formation in vivo, these observations support a model whereby PomX$^{WT}$ serves as a scaffold protein for cluster formation in vivo and in which PomX$^C$ has a key role in this scaffolding function by self-interacting and interacting with PomY. The observation that neither PomX$^N$ nor PomX$^C$ alone is sufficient to form a cluster in vivo suggests that these two domains may also interact. Our experiments suggest that these interactions are of low affinity because there were not detected by any of the methods used here (BACTH, TEM, high-speed centrifugation, and pull-down experiments). Alternatively, the conformation of the two separated domains could be different from that in PomX$^{WT}$, and, therefore, no interactions were detected. Altogether, we suggest that PomX$^{WT}$ monomers in vivo interact via their coiled-coil domain in PomX$^C$ to spontaneously form a polymeric structure that is stabilized or modified by PomX$^N$; this polymeric structure, in turn, serves as a scaffold to recruit PomY resulting in formation of the PomX/Y complex. PomZ in its ATP-bound dimeric form is recruited to this PomX/Y complex and associates it to the nucleoid. Because PomX$^{WT}$ and the likely unstructured PomX$^N$ stimulate PomZ ATPase activity to the same extent, these observations also strongly support that PomX$^{WT}$ monomers in the polymeric structure in vitro and cluster in vivo stimulate PomZ ATPase

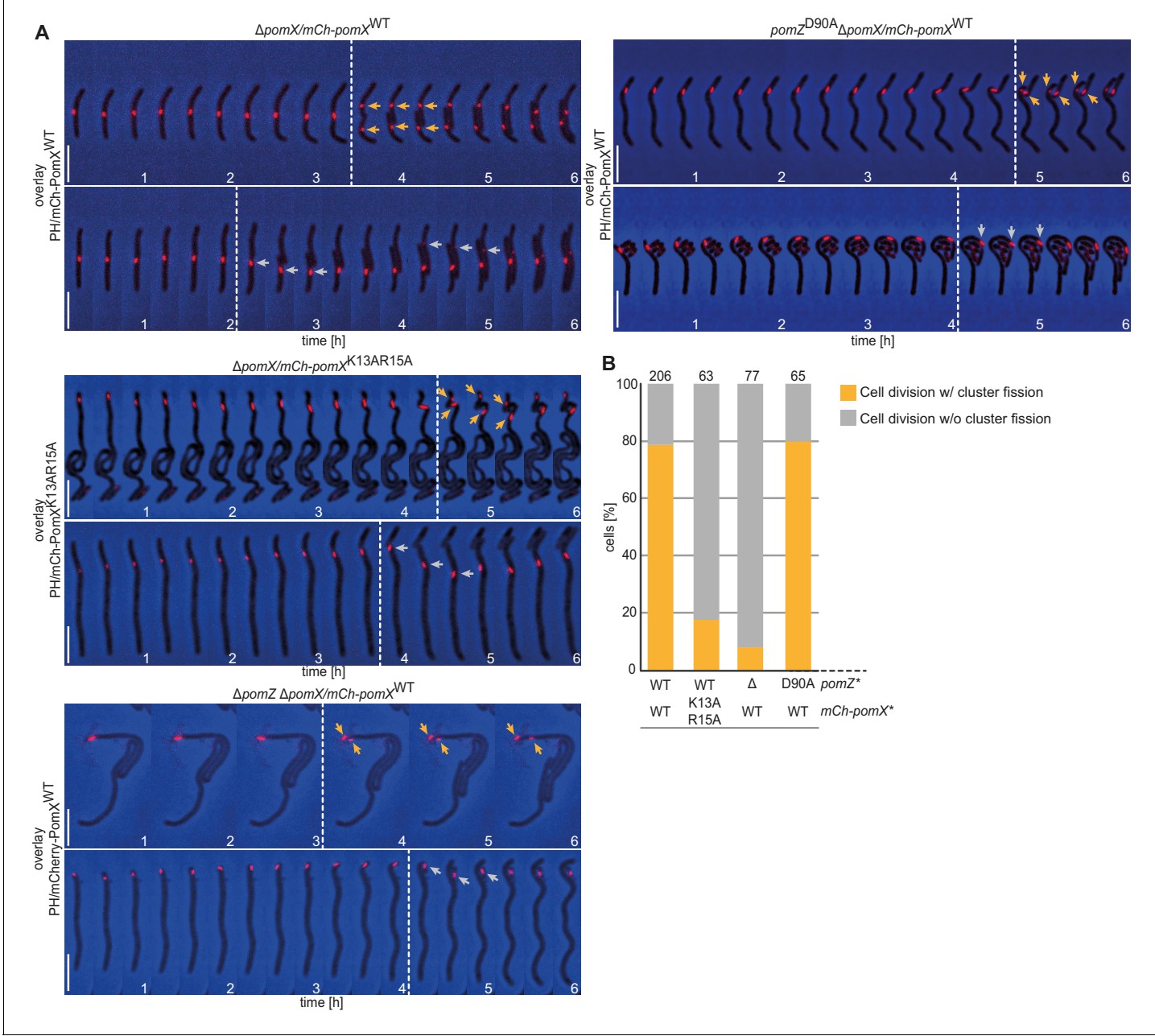

**Figure 6.** The PomX/PomZ interaction is important for cluster fission during division. (**A**) Fluorescence time-lapse microscopy of mCh-PomX variants in cells of indicated genotypes. Overlays of representative mCh images and PH are shown in 20 min intervals. Stippled lines indicate cell division events. Orange and gray arrows mark mCh-PomX clusters in daughter cells after cell division with cluster fission and without cluster fission, respectively. Scale bar, 5 μm. (**B**) Quantification of cluster fission during cell division in cells of indicated genotypes. Cell division events were divided into those with (orange) and without (gray) cluster fission. Number of analyzed cell divisions is shown on top. The same results were obtained in two independent experiments.

The online version of this article includes the following source data for figure 6:

**Source data 1.** Source data for *Figure 6B*.

activity independently of each other. In vitro, purified PomX$^{WT}$ spontaneously polymerizes under all conditions tested. We speculate that this spontaneous polymerization ensures that one, and only one, PomX scaffold is formed per cell in vivo, thus guaranteeing that only one PomX/Y/Z complex is formed per cell as would be required for a complex that defines the site of cell division.

By analyzing PomZ variants, we previously showed that PomZ ATPase activity is essential for PomX/Y/Z cluster translocation and cluster localization to midcell. Because the PomX variants with reduced AAP activity resulted in the formation of PomX/Y/Z clusters that were typically not at midcell, we conclude that the low intrinsic PomZ ATPase activity is not sufficient to fuel translocation of the PomX/Y/Z cluster to midcell and that this translocation is fueled by PomX, and likely also by PomY, stimulated ATP hydrolysis by PomZ. In the PomX AAP mutants, constrictions were formed at WT frequencies over the PomX/Y/Z cluster. Because these clusters were typically not at midcell, these PomX variants resulted in divisions away from midcell giving rise to filamentous cells and minicells. Thus, PomX AAP mutants are fully competent in stimulating cell division but deficient in positioning the PomX/Y/Z cluster at midcell. These observations also support that PomX-stimulated PomZ ATPase activity is not important for stimulation of Z-ring formation. In cells lacking the PomX protein, PomY and PomZ do not form clusters and cell divisions occur at a low frequency (*Schumacher et al., 2017*); by contrast, the PomX AAP mutants still support PomX/Y/Z cluster formation and cell division. Thus, PomX AAP mutants retain partial PomX activity. Importantly, in mutants lacking PomZ, the PomX/Y cluster is also mostly away from midcell, and Z-rings and constrictions are formed over the cluster away from midcell, but at a much-reduced frequency compared to WT. Thus, PomX AAP mutants and a mutant lacking PomZ phenocopy each other with respect to PomX/Y/Z cluster positioning but not with respect to constriction frequency. We suggest that this difference is the result of different cluster compositions, that is the Pom clusters formed in the AAP mutants contain PomZ as well as PomY, which both interact with FtsZ (*Schumacher et al., 2017*), while the Pom clusters formed in the absence of PomZ only contain PomX and PomY. Thus, the division defects in Δ*pom* mutants are a consequence of reduced formation and mislocalization of division constrictions, while the PomX AAP mutants are only deficient in cell division localization.

DNA binding by PomZ is important for the low intrinsic as well as PomX-stimulated ATPase activity. This is similar to what has been described for several DNA-binding ParA ATPases and their partner AAPs (*Ah-Seng et al., 2009*; *Kiekebusch et al., 2012*; *Leonard et al., 2005*; *Lim et al., 2014*; *Scholefield et al., 2011*; *Schumacher et al., 2017*). Similarly, MinE-dependent stimulation of ATP hydrolysis by a MinD variant that does not bind the membrane is reduced (*Hu and Lutkenhaus, 2003*). However, it remains unknown how DNA or membrane binding makes these ATPases competent for ATPase activity. Likewise, the precise molecular mechanism by which AAPs stimulate ATPase activity of their cognate ParA/MinD family ATPase remains unknown. However, in the case of the *E. coli* MinD/MinE system, it has been suggested that MinD ATPase activation by MinE involves the asymmetric interaction of the N-terminus of a MinE monomer with the ATP-bound MinD dimer (*Park et al., 2012*). Our data support that activation of many ParA/MinD family ATPases by their partner AAP may involve a shared mechanism: In agreement with the findings here that PomX[NPEP] is enriched in positively charged residues and that Lys13 and Arg15 are essential for PomX AAP activity, it has previously been noted that ParB-type AAPs, MinE-type AAPs and the non-homologous AAP ParG of plasmid TP228 that functions together with the ParA ATPase ParF contain a stretch of N-terminal residues rich in positively charged amino acids (*Barillà et al., 2007*; *Leonard et al., 2005*; *Ah-Seng et al., 2009*; *Ghasriani et al., 2010*; *Figure 7*). For several of these AAPs, it has been shown that one or more of the positively charged residues are important for AAP activity (*Figure 7*; *Leonard et al., 2005*; *Scholefield et al., 2011*; *Barillà et al., 2007*; *Ah-Seng et al., 2009*; *Park et al., 2012*; *Hu and Lutkenhaus, 2001*; *Ghasriani et al., 2010*). Also, as noted above, in the case of Spo0J of *T. thermophilus*, plasmid F SopB, and MinE of *N. gonorrhoeae*, 20, 52, and 22 N-terminal residues are sufficient for AAP activity. Thus, PomX is the fourth type of ParA/MinD AAP that displays this characteristic N-terminus and in which positively charged residues are important for AAP activity. Intriguingly, TlpT, which is the suggested AAP of the ParA-like ATPase PpfA involved in translocation and positioning of the large cytoplasmic chemoreceptor cluster in *Rhodobacter sphaeroides* (*Roberts et al., 2012*), and McdB, which is the AAP of the ParA-like ATPase McdA important for carboxysome translocation and positioning in *Synechococcus elongatus* (*MacCready et al., 2018*) both have an N-terminal extension enriched in positively charged residues (*Figure 7*). Altogether, these findings lend further support to the notion that many AAPs of ParA/MinD ATPases use the same mechanism to stimulate ATPase activity (*Leonard et al., 2005*; *Park et al., 2011*; *Zhang and Schumacher, 2017*; *Barillà et al., 2007*). Moreover, they support that ParA/MinD AAPs display remarkable plasticity and modularity in which a stretch of N-terminal amino acids enriched in positively charged residues grafted onto an interaction domain, which is involved

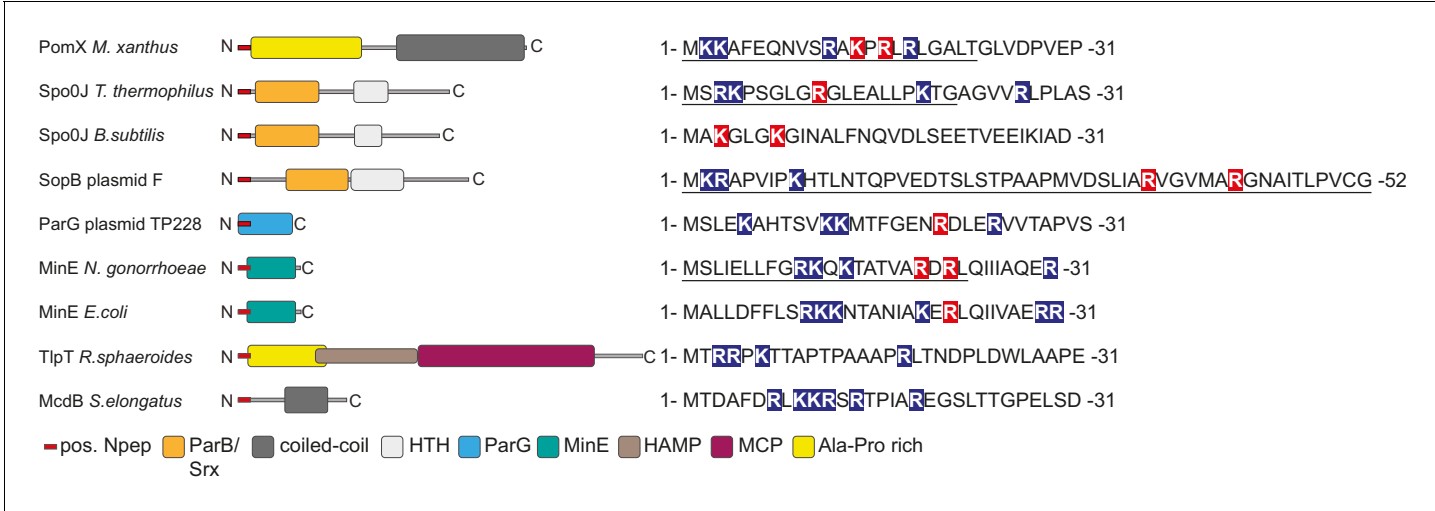

**Figure 7.** AAPs of MinD/ParA ATPases are diverse but share common features. Left, domain analysis of known and predicted AAPs of ParA/MinD ATPases with key below. Sequences on the right, N-terminus of indicated proteins. Positively charged amino acids are indicated on blue, and positively charged residues experimentally demonstrated to be important for AAP activity on red. Underlined sequences indicate peptides experimentally demonstrated to have AAP activity. Spo0J of *T. thermophilus* and *B. subtilis,* and SopB of plasmid F are ParB homologs.

in protein-protein, protein-DNA, or protein-membrane interaction, can generate an AAP. Interestingly, PomY, the second AAP of PomZ (*Schumacher et al., 2017*), does not have an N-terminus enriched in positively charged residues suggesting that its mode of action could be different from that of previously described AAPs.

In addition to similarities at the biochemical level, systems incorporating a DNA-binding ParA ATPase and an AAP(s) also share similarities in their translocation. The PomX/Y complex, ParB-*parS* complexes, cytoplasmic chemoreceptor clusters and carboxysomes are all large structures that are translocated as cargo on the nucleoid in a mechanism that depends on an AAP(s) stimulating ATP hydrolysis by the cognate DNA-binding ParA ATPase. Importantly, the AAP(s) is an integral part of the translocated cargo. We speculate that the integration of the AAP activity into a large cargo structure provides an elegant solution to guarantee that ATPase activation is spatially restricted to the transported cargo.

In addition to serving as a scaffold for PomX/Y/Z cluster formation and as a PomZ AAP, we report that PomX has a third important function in PomX/Y/Z cluster fission during division. In the presence of PomX$^{WT}$, the cluster visibly undergoes fission during ~80% of divisions in otherwise WT cells, and those cells that do not receive a cluster slowly rebuild a cluster [here; (*Schumacher et al., 2017*)]. As a result, most WT cells contain a visible PomX/Y/Z cluster. By contrast, a PomX AAP mutant did not support fission as efficiently as PomX$^{WT}$ and fewer cells contain a cluster. We speculate that the reduced frequency of cluster fission events contributes to the cell division defect in the PomX AAP mutants. In cells lacking PomZ, the PomX/Y cluster also does not split efficiently during division, while the ATP-locked dimeric PomZ$^{D90A}$ stimulates fission as efficiently as PomZ$^{WT}$. In all these mutants, division occurs over the Pom cluster. Additionally, in cells treated with cephalexin, which blocks cell division, cluster fission also does not occur (*Schumacher et al., 2017*; *Treuner-Lange et al., 2013*). Because cephalexin-treated cells still segregate their chromosomes (*Schumacher et al., 2017*; *Treuner-Lange et al., 2013*), these observations support that cell division is essential for fission, chromosome segregation is not sufficient, and the PomX/Z interaction is important while PomZ ATPase activity is not. This is in contrast to the SopB/SopA system for plasmid F segregation. For this system, it has been shown that SopB-stimulated ATPase activity of SopA is important to split plasmid clusters (*Ah-Seng et al., 2013*) suggesting that the splitting of plasmid clusters and PomX/Y/Z cluster fission may occur by different mechanisms. In the future, it will be interesting to establish the mechanism for PomX/Y/Z cluster fission in detail.

## Materials and methods

### *M. xanthus* and *E. coli* strains and growth

Strains, plasmids, and primers are listed in the Key Resources Table. *M. xanthus* strains are derivatives of DK1622 WT (*Kaiser, 1979*). *M. xanthus* strains were cultivated in 1% CTT medium (1% casitone, 10 mM Tris-HCl pH 7.6, 1 mM KPO$_4$ pH 7.6, 8 mM MgSO$_4$) or on 1% CTT 1.5% agar plates (*Hodgkin and Kaiser, 1977*). Kanamycin, oxytetracycline, and gentamycin were added at concentrations of 50 µg/ml, 10 µg/ml, and 10 µg/ml, respectively. Growth was measured as an increase in optical density (OD) at 550 nm. *M. xanthus* cells were transformed by electroporation. In-frame deletions were generated as described (*Shi et al., 2008*). Plasmids were integrated by site-specific recombination at the Mx8 *attB* locus or by homologous recombination at the endogenous site. All plasmids were verified by sequencing. All strains were verified by PCR. Non-motile strains were generated to allow time-lapse microscopy for several hours by deletion of *mglA* (*Miertzschke et al., 2011*; *Schumacher and Søgaard-Andersen, 2018*). *E. coli* strains were grown in LB or 2xYT medium in the presence of relevant antibiotics or on LB plates containing 1.5% agar (*Sambrook and Russell, 2001*). Plasmids were propagated in *E. coli* NEB Turbo cells (New England Biolabs) (F′ *proA*⁺*B*⁺ *lacI*$^q$ Δ*lacZM15/fhuA2* Δ*(lac-proAB)* *glnV galK16 galE15 R(zgb-210::Tn10)* Tet$^S$ *endA1 thi-1* Δ*(hsdS-mcrB)* 5). Growth of *E. coli* was measured as an increase in OD at 600 nm.

### Fluorescence microscopy and live cell imaging

Fluorescence microscopy was performed as described (*Schumacher et al., 2017*). Briefly, exponentially growing cells were transferred to slides with a thin 1.0% agarose pad (SeaKem LE agarose, Cambrex) with TPM buffer (10 mM Tris-HCl pH 7.6, 1 mM KH$_2$PO$_4$ pH 7.6, 8 mM MgSO$_4$), covered with a coverslip and imaged using a temperature-controlled Leica DMi6000B inverted microscope with an HCX PL FLUOTAR objective at 32°C. Phase-contrast and fluorescence images were recorded with a Hamamatsu ORCA-flash 4.0 sCMOS camera using the LASX software (Leica Microsystems). Time-lapse microscopy was performed as described (*Schumacher and Søgaard-Andersen, 2018*). Briefly, cells were transferred to a coverslip mounted on a metallic microscopy slide and covered with a pre-warmed 1% agarose pad supplemented with 0.2% casitone in TPM buffer. Slides were covered with parafilm to retain humidity of the agarose. Live-cell imaging was performed at 32°C. For DNA staining, cells were incubated with 1 mg/ml 2-(4-Amidinophenyl)−6-indolcarbamidine-dihydrochloride (DAPI) for 10 min at 32°C before microscopy. Image processing was performed with Metamorph_v 7.5 (Molecular Devices). For image analysis, cellular outlines were obtained from phase-contrast images using Oufti and manually corrected if necessary (*Paintdakhi et al., 2016*). Fluorescence microscopy image analysis was performed with a custom-made Matlab script (Matlab R2018a, MathWorks) available at https://github.com/SBergeler/ImageAnalysisMyxo copy archived at 500 cells from a dataset (unless otherwise stated) and calculating the length of the cells' center-lines (using meshes obtained from Oufti). Next, the fluorescence intensity of each cell was corrected by the background fluorescence locally around each cell. Cells were ordered by length and oriented according to the cell segment with the brightest intensity. The brightest 3% of the intensity values of all cells were set to the maximal intensity value, which is scaled to one, to be able to visualize fluorescence intensity variations inside the cells despite very bright clusters.

### Bacterial two hybrid assay

BACTH experiments were performed as described (*Karimova et al., 1998*). Relevant genes were cloned into the appropriate vectors to construct N-terminal and C-terminal fusions with the 25 kDa N-terminal or the 18 kDa C-terminal adenylate cyclase fragments of *B. pertussis*. Restoration of cAMP production was observed by the formation of blue color on LB agar supplemented with 80 µg/ml 5-bromo-4-chloro-3-indolyl-β-D-galactopyranoside (X-Gal) and 0.25 mM isopropyl-β-D-thiogalactoside (IPTG). As positive control, the leucine zipper from GCN4 was fused to the T18 and the T25 fragment. As a negative control, plasmids were co-transformed that only expressed the T18 or T25 fragment. All tested interactions were spotted on the same LB agar plates with positive control and all corresponding negative controls.

## Western blot analysis

Western blot analyses were performed as described (*Sambrook and Russell, 2001*) with rabbit polyclonal $\alpha$-PomX (1:15000), $\alpha$-PomY (1:15000), $\alpha$-PomZ (1:10000) (*Schumacher et al., 2017*), $\alpha$-PilC (1:3000) (*Bulyha et al., 2009*), or $\alpha$-mCh (1:10000; Biovision) primary antibodies together with horseradish-conjugated goat $\alpha$-rabbit immunoglobulin G (Sigma-Aldrich) as secondary antibody (1:25000). Blots were developed using Luminata Forte Western HRP Substrate (Millipore) and visualized using a LAS-4000 luminescent image analyzer (Fujifilm).

## Protein purification

$His_6$-PomZ was purified from *E. coli* as described (*Treuner-Lange et al., 2013*). $PomX^{WT}$-$His_6$ and PomY-$His_6$ were purified as described (*Schumacher et al., 2017*) using plasmids pEMR3 and pEMR1, respectively. $PomX^{K13AR15A}$-$His_6$ was purified as PomX-$His_6$. Briefly, plasmid pSH58 was propagated in *E. coli* NiCo21(DE3) cells (NEB), grown in LB medium with 50 µg/ml kanamycin at 30°C to an $OD_{600}$ of 0.6–0.7. Protein expression was induced with 0.4 mM IPTG for 16 hr at 18°C. Cells were harvested by centrifugation at 6000 g for 20 min at 4°C. Cells were washed with lysis buffer 1 (50mM $NaH_2PO_4$; 300 mM NaCl; 10 mM imidazole; pH 8.0 (adjusted with NaOH)) and lysed in 50 ml lysis buffer 2 (lysis buffer 1 with 0.1 mM EDTA; 1 mM β-mercaptoethanol; 100 mg/ml phenylmethylsulfonyl fluoride (PMSF); 1× complete protease inhibitor (Roche Diagnostics GmbH); 10 U/ml DNase 1) by sonication in three rounds of sonication for 5 min with a Branson Sonifier (Duty cycle 4; output control 40%) (Heinemann) on ice. Cell debris was removed by centrifugation at 4700 g for 45 min at 4°C. $PomX^{K13AR15A}$-$His_6$ was affinity purified with Protino Ni-NTA resin (Macherey-Nagel) from a batch, equilibrated in lysis buffer 1. $PomX^{K13AR15A}$-$His_6$ was eluted from the resin by washing 1× with 5 ml elution buffer 1 (lysis buffer 1 with 50 mM imidazole) and 3 x with 5 ml elution buffer 2 (lysis buffer 1 with 250 mM imidazole). Purified $PomX^{K13AR15A}$-$His_6$ was dialyzed 4× against 2 l dialysis buffer (50 mM Hepes/NaOH pH 7.2; 50 mM KCl; 0.1 mM EDTA; 1 mM β-mercaptoethanol; 10% (v/v) glycerol). Proteins were frozen in liquid nitrogen and stored at −80°C until used.

To purify $PomX^N$-$His_6$ plasmid pAH157 was propagated in *E. coli* NiCo21(DE3) cells, grown in 2×YT medium with 50 µg/ml kanamycin and 0.5% glucose at 30°C to an $OD_{600}$ of 0.6–0.7. Protein expression was induced with 0.4 mM IPTG for 16 hr at 18°C. Cells were harvested and lysed as described for $PomX^{K13AR15A}$. $PomX^N$-$His_6$ was affinity purified with Protino Ni-NTA resin (Macherey-Nagel) from a batch, equilibrated in lysis buffer 1. Contaminating proteins were eluted from the resin by washing 6× with 40 ml wash buffer 1 (lysis buffer 1 with 20 mM imidazole), 1× with 40 ml wash buffer 2 (lysis buffer 1 with 50 mM imidazole). $PomX^N$-$His_6$ was eluted from the resin by washing with 1 × 10 ml elution buffer 1 (lysis buffer 1 with 100 mM imidazole), 1× with 10 ml elution buffer 2 (lysis buffer 1 with 150 mM imidazole) and 1× with 10 ml elution buffer 3 (lysis buffer 1 with 200 mM imidazole). The elution fractions were pooled and loaded onto a HiLoad 16/600 Superdex 200 pg gel filtration column (GE Healthcare) that was equilibrated with dialysis buffer. Elution fractions were pooled. Proteins were frozen in liquid nitrogen and stored at −80°C until used.

For purification of $PomX^{N\_K13AR15A}$-$His_6$ plasmid pAH165 was propagated in *E. coli* Rosetta2(DE3) cells, grown in 2×YT medium with 50 µg/ml kanamycin, 30 µg/ml chloramphenicol and 0.5% glucose at 30°C to an $OD_{600}$ of 0.6–0.7. Protein expression was induced with 0.5 mM IPTG for 16 hr at 18°C. Cells were harvested and lysed as described for $PomX^{K13AR15A}$. $PomX^{N\_K13AR15A}$-$His_6$ was purified from cleared lysates with a 5 ml HiTrap Chelating HP column loaded with $NiSO_4$ and equilibrated with lysis buffer 1. The column was washed with 20 column volumes (CVs) lysis buffer 1. The protein was eluted with elution buffer (50 mM $NaH_2PO_4$; 300 mM NaCl; 500 mM imidazole; pH 8.0 [adjusted with NaOH]) with a gradient of 20 CV. Fractions containing $PomX^{N\_K13AR15A}$-$His_6$ were pooled and concentrated with an Amicon Ultra-15 centrifugation filter device with a cutoff of 3 kDa and loaded onto a HiLoad 16/600 Superdex 200 pg gel filtration column (GE Healthcare) that was equilibrated with dialysis buffer. Elution fractions were pooled. Proteins were frozen in liquid nitrogen and stored at −80°C until used.

To purify $PomX^C$-$His_6$ plasmid pAH152 was propagated in *E. coli* Rosetta2(DE3) cells, grown in 2×YT medium with 50 µg/ml kanamycin, 30 µg/ml chloramphenicol and 0.5% glucose at 37°C to an $OD_{600}$ of 0.6–0.7. Protein expression was induced with 0.5 mM IPTG for 4 hr at 37°C. Cells were harvested and lysed as described for $PomX^{K13AR15A}$. $PomX^C$-$His_6$ was affinity purified with Protino Ni-NTA resin (Macherey-Nagel) from a batch, equilibrated with lysis buffer 1. Contaminating proteins

were eluted from the resin by washing 6× with 40 ml wash buffer 1 (lysis buffer 1 with 20 mM imidazole), 1× with 40 ml wash buffer 2 (lysis buffer 1 with 50 mM imidazole) and 2× with 40 ml wash buffer 3 (lysis buffer 1 with 100 mM imidazole) and 1× with 40 ml wash buffer 4 (lysis buffer 1 with 150 mM imidazole). PomX$^C$-His$_6$ was eluted from the resin with 3 × 10 ml elution buffer (lysis buffer 1 with 250 mM imidazole). The elution fractions were pooled and dialyzed against 4 × 2 l of dialysis buffer at 4°C. Proteins were frozen in liquid nitrogen and stored at −80°C until used.

For purification of PomX$^N$-Strep, plasmid pDS232 was propagated in *E. coli* Rosetta2(DE3) cells, grown in LB medium with 50 µg/ml kanamycin, 30 µg/ml chloramphenicol and 0.5% glucose at 32°C to an OD$_{600}$ of 0.6–0.7. Protein expression was induced with 0.5 mM IPTG for 2 hr at 32°C. Cells were harvested and lysed as described for PomX$^{K13AR15A}$-His$_6$ but in StrepTag lysis buffer (100 mM Tris-HCl pH 8.0, 150 mM NaCl, 1 mM EDTA, 1 mM dithiothreitol (DTT)). PomX$^N$-Strep was purified from cleared lysates with a 5 ml StrepTrap HP column equilibrated with StrepTag lysis buffer. The column was washed with 20 CV StrepTag lysis buffer. The protein was eluted with StrepTag elution buffer (StrepTag lysis buffer with 2.5 mM D-desthiobiotin). Fractions containing PomX$^N$-Strep were pooled and dialyzed against 2 × 3 l of dialysis buffer at 4°C. Proteins were frozen in liquid nitrogen and stored at −80°C until used.

For purification of PomX$^C$-Strep, plasmid pDS333 was propagated in *E. coli* Rosetta2(DE3) cells, grown in LB medium with 50 µg/ml kanamycin, 30 µg/ml chloramphenicol and 0.5% glucose at 32°C to an OD$_{600}$ of 0.6–0.7. Protein expression was induced with 1 mM IPTG for 18 hr at 18°C. Cells were harvested and lysed as described for PomX$^N$-Strep. PomX$^C$-Strep was purified from a batch using 2 ml Strep-TactinXT 4Flow resin (iba), equilibrated with StrepTag lysis buffer. The resin was incubated with the cleared lysate for 1 hr at 4°C on a rotary shaker. Contaminating proteins were eluted from the resin by washing 5× with 10 ml StrepTag lysis buffer. The protein was eluted with 1× BXT buffer (100 mM Tris-HCl pH8.0, 150 mM NaCl, 1 mM EDTA, 50 mM biotin) (iba). Fractions containing PomX$^C$-Strep were pooled and dialyzed against 2 × 5 l of dialysis buffer at 4°C. Proteins were frozen in liquid nitrogen and stored at −80°C until used.

## Protein sedimentation assay

Before sedimentation experiments, a clearing spin was performed for all proteins to be analyzed at 20,000 *g* for 10 min at 4°C. Proteins at a final concentration of 3 µM in a total volume of 50 µl were mixed and incubated for 1 hr at 32°C in a buffer (50 mM Hepes/NaOH, pH 7.2, 50 mM KCl, 1 mM β-mercaptoethanol, 10 mM MgCl$_2$). Samples were separated into soluble and insoluble fractions by high-speed centrifugation (160,000 *g*, 60 min, 25°C). Insoluble and soluble fractions were separated, and volumes adjusted with 1× SDS sample buffer. Fractions were separated by SDS-PAGE and stained with Instant Blue (expedion) for 10 min.

## In vitro pull-down experiments

Ten µM protein alone or pre-mixed as indicated were incubated for 1 hr at 32°C in reaction buffer (50 mM HEPES/NaOH pH 7.2, 50 mM KCl, 10 mM MgCl$_2$) in a total volume of 200 µl and applied to 20 µl 5% (v/v) MagStrepXT beads (iba) for 30 min. Magnetic beads were washed 10× with 200 µl reaction buffer. Proteins were eluted with 200 µl 1× BXT buffer (100 mM Tris-HCl pH8.0, 150 mM NaCl, 1 mM EDTA, 50 mM biotin) (iba). 10 µl per sample were separated by SDS-PAGE.

## Negative stain transmission electron microscopy

To fix and stain protein samples for negative stain TEM, 10 µl of a protein sample of interest (protein concentration before application onto the EM grid 3 µM) was applied onto an EM grid (Plano) and incubated for 1 min at 25°C. Residual liquid was blotted through the grid by applying the grid's unused side on Whatman paper. The grid was washed twice with double-distilled H$_2$O. For staining, 10 µl of 1% uranyl acetate solution was applied onto the grid for 1 min and blotted through with a Whatman paper. If protein mixtures were applied to the EM grid, proteins of interest at a concentration of 3 µM were pre-mixed in a low-binding microtube (Sarstedt) and incubated for 10 min at 25°C before application onto the EM grid. Finished grids were stored in a grid holder for several months at room temperature. Electron microscopy was performed with a CM120 electron microscope (FEI) at 120kV.

## ATPase assay

ATP hydrolysis was determined using a 96-well NADH-coupled enzymatic assay (*Kiianitsa et al., 2003*) with modifications. Protein concentration was determined using Protein Assay Dye Reagent Concentrate (BioRad). Assays were performed in reaction buffer (50 mM HEPES/NaOH pH 7.2, 50 mM KCl, 10 mM $MgCl_2$) with 0.5 mM nicotinamide adenine dinucleotide (NADH) and 2 mM phosphoenolpyruvate and 3 µl of a pyruvate kinase/lactate dehydrogenase mix (PYK/LDH; Sigma). $PomX^{NPEP}$ and $PomX^{NPEP\_K13AR15A}$ peptides (MKKAFEQNVSRA**KPR**LRLGALT and MKKAFEQNVS-RA**APA**LRLGALT) were purchased from Thermo Scientific. If appropriate, herring sperm DNA was added at a concentration of 60 µg/ml unless otherwise stated. Buffer was pre-mixed with proteins in low-binding microtubes (Sarstedt) on ice. To correct for glycerol in the assays, dialysis buffer was added if necessary. A total of 100 µl mixtures were transferred into transparent UV-STAR µCLEAR 96-well microplates (Greiner bio-one). The reaction was started by the addition of 1 mM ATP. Measurements were performed in an infinite M200PRO (Tecan) for 2 hr in 30 s intervals at 32°C shaking at 340 nm wavelength. To account for background by spontaneous ATP hydrolysis and UV-induced NADH decomposition, all assays were performed without the addition of $His_6$-PomZ and measurements were subtracted. The light path was determined experimentally with known NADH concentrations to be 0.248 cm. The extinction coefficient of NADH $\varepsilon340 = 6220 \ M^{-1}cm^{-1}$ was used.

## Analytical size-exclusion chromatography

Experiments were carried out in dialysis buffer (50 mM Hepes/NaOH pH 7.2; 50 mM KCl; 0.1 mM EDTA; 1 mM β-mercaptoethanol; 10% (v/v) glycerol). $PomX^N$-$His_6$ and $PomX^{N\_K13AR15A}$-$His_6$ were applied onto a Superdex 200 10/300 GL gel filtration column equilibrated with dialysis buffer. Blue dextran (2000 kDa), ferritin (440 kDa), conalbumin (75 kDa), ovalbumin (43 kDa), carbonic anhydrase (29 kDa), RNAse A (13.7 kDa), and aprotinin (6.5 kDa) were used as standards with the same buffer conditions to calibrate the column.

## Bioinformatics

Gene and protein sequences of PomX, PomY, and PomZ were obtained from NCBI. PomX homologs were identified in a best-best hit reciprocal BlastP analysis from fully-sequenced genomes of Myxobacteria (*Altschul et al., 1990*). The similarity and identity of proteins were calculated from pairwise sequence alignments with EMBOSS Needle (*Madeira et al., 2019*). Domain analyses were performed with SMART (*Letunic and Bork, 2018*), PROSITE and Pfam (*El-Gebali et al., 2019*). Multiple sequence alignments were created with MUSCLE (*Madeira et al., 2019*) and further edited with Bioedit (https://bioedit.software.informer.com/7.2/). Consensus sequences of multiple sequence protein alignments were created with Weblogo 3 (*Crooks et al., 2004*). Proteins used: PomXMx (ABF89666; MXAN_0636), PomXMm (ATB45064; MYMAC_000648), PomXMh (AKQ69458; A176_006370), PomXMf (AKF79435; MFUL124B02_03625), PomXMs (AGC41991; MYSTI_00641), PomXCc (AFE03552; COCOR_00544), PomXMb (ATB30647; MEBOL_004108), PomXCf (ATB35470; CYFUS_000883), PomXAg (AKI99966; AA314_01593), PomXSa (ADO75429; STAUR_7674), PomXVi (AKU92216; AKJ08_2603), PomXAd (ABC83600; Adeh_3834). Spo0J Tt (AAS81946.1), Spo0J Bs (P26497.2) ParG TP228 Ec (WP_139578510.1), SopB F Ec (BAA97917.1), MinE Ng (AAK30127.1), MinE Ec (EFB7450413.1), TlpT Rs (ABA79218.1), McdB Se (ABB57864.1).

## Statistics

The mean and standard deviation (STDEV) were calculated with Excel 2016. Localization patterns from fluorescence microscopy data were quantified based on the indicated n-value per strain. Boxplots were generated with SigmaPlot 14.0 (Systat). Statistical analysis was performed with SigmaPlot 14.0. All data sets were tested for normality using a Shapiro-Wilk test. For data with a non-normal distribution, a Mann-Whitney test was applied to test for significant differences.

## Acknowledgements

We thank Sabrina Huneke-Vogt for assistance with plasmid constructions, Manon Wigbers for help with the image analysis script, and Anke Treuner-Lange for many helpful discussions.

## Additional information

### Funding

| Funder | Grant reference number | Author |
|---|---|---|
| Deutsche Forschungsge-meinschaft | TRR 174 | Erwin Frey<br>Lotte Søgaard-Andersen |
| Max Planck Society | | Lotte Søgaard-Andersen |

The funders had no role in study design, data collection and interpretation, or the decision to submit the work for publication.

### Author contributions

Dominik Schumacher, Conceptualization, Resources, Data curation, Formal analysis, Validation, Investigation, Visualization, Methodology, Writing - original draft, Project administration, Writing - review and editing; Andrea Harms, Data curation, Validation, Investigation; Silke Bergeler, Software, Methodology, Writing - review and editing; Erwin Frey, Supervision, Funding acquisition, Writing - review and editing; Lotte Søgaard-Andersen, Conceptualization, Supervision, Funding acquisition, Writing - original draft, Project administration, Writing - review and editing

### Author ORCIDs

Erwin Frey (iD) http://orcid.org/0000-0001-8792-3358
Lotte Søgaard-Andersen (iD) https://orcid.org/0000-0002-0674-0013

### Decision letter and Author response

Decision letter https://doi.org/10.7554/eLife.66160.sa1
Author response https://doi.org/10.7554/eLife.66160.sa2

## Additional files

### Supplementary files

• Transparent reporting form

### Data availability

All data supporting this study are available within the article and supporting files. Source data files have been provided for Figures 1, 2, 3, 4, 5 and 6.

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

# Appendix 1

**Appendix 1—key resources table**

| Reagent type (species) or resource | Designation | Source or reference | Identifiers | Additional information |
|---|---|---|---|---|
| Gene (*M. xanthus*) | *pomX* | NCBI | *mxan_0636* new locus tag MXAN_RS03090 | |
| Gene (*M. xanthus*) | *pomY* | NCBI | *mxan_0634* new locus tag MXAN_RS03080 | |
| Gene (*M. xanthus*) | *pomZ* | NCBI | *mxan_0635* new locus tag MXAN_RS03085 | |
| Strain, strain background (*E. coli*) | Arctic Express DE3 RP | Agilent Technologies | *E. coli* B F⁻ *ompT hsdS*(r⁻_B m⁻_B) *dcm*⁺Tet^R *gal* λ(DE3) *endA* Hte [*cpn10 cpn60* Gent^R] | Used for protein expression |
| Strain, strain background (*E. coli*) | Rosetta 2 DE3 | Novagen | F⁻ *ompT hsdS*_B(r⁻_B m⁻_B) *gal dcm* (DE3) pRARE2 (Cam^R) | Used for protein expression |
| Strain, strain background (*E. coli*) | NiCo21(DE3) | New England Biolabs | *can::CBD fhuA2 [lon] ompT gal (λ DE3) [dcm] arnA::CBD slyD::CBD glmS6Ala ΔhsdS λ DE3 = λ sBamHIo ΔEcoRI-B int:: (lacI::PlacUV5::T7 gene1) i21 Δnin5* | Used for protein expression |
| Strain, strain background (*E. coli*) | NEB Turbo | New England Biolabs | F' *proA⁺B⁺ lacI^q(ΔlacZM15/fhuA2 Δ(lac-proAB) glnV galK16 galE15 R(zgb-210::Tn10) Tet^S endA1 thi-1 Δ(hsdS-mcrB)5)* | Used for cloning |
| Strain, strain background (*M. xanthus*) | DK1622 | DOI: 10.1073/pnas.76.11.5952 | Wildtype | |
| Strain, strain background (*M. xanthus*) | SA3108 | DOI: 10.1111/mmi.12094 | ΔpomZ | Strain with an in-frame deletion in *pomZ* |
| Strain, strain background (*M. xanthus*) | SA3146 | DOI: 10.1111/mmi.12094 | ΔpomZ; attB::P_mxan0635 pomZ^D90A-mCh (pKA43) | Strain expressing PomZ^D90A-mCh in a ΔpomZ background |
| Strain, strain background (*M. xanthus*) | SA4223 | https://doi.org/10.1016/j.devcel.2017.04.011 | ΔpomX | Strain with an in-frame deletion in *pomX* |
| Strain, strain background (*M. xanthus*) | SA4252 | https://doi.org/10.1016/j.devcel.2017.04.011 | ΔpomX; attB::P_mxan0635 mCh-pomX (pAH53) | Strain expressing mCh-PomX in a ΔpomX background |
| Strain, strain background (*M. xanthus*) | SA4703 | https://doi.org/10.1016/j.devcel.2017.04.011 | ΔpomY | Strain with an in-frame deletion in *pomY* |

*Continued on next page*

*Appendix 1—key resources table continued*

| Reagent type (species) or resource | Designation | Source or reference | Identifiers | Additional information |
|---|---|---|---|---|
| Strain, strain background (*M. xanthus*) | SA4712 | https://doi.org/10.1016/j.devcel.2017.04.011 | $\Delta pomY$; $attB::P_{pilA}\ pomY\text{-}mCh$ (pDS7) | Strain expressing PomY-mCh in a $\Delta pomY$ background |
| Strain, strain background (*M. xanthus*) | SA4797 | https://doi.org/10.1016/j.devcel.2017.04.011 | $\Delta mglA$; $\Delta pomX$; $attB::P_{mxan0635}\ mCh\text{-}pomX$ (pAH53) | Strain expressing mCh-PomX in a non-motile $\Delta pomX$ background |
| Strain, strain background (*M. xanthus*) | SA4297 | this study | Wild-type; $attB::P_{mxan0635}\ mCh\text{-}pomX$ (pAH53) | Strain expressing mCh-PomX in WT background |
| Strain, strain background (*M. xanthus*) | SA6100 | this study | $pomX::pomX^{K13AR15A}$ | Strain with a replacement of *pomX* with the $pomX^{K13AR15A}$ allele |
| Strain, strain background (*M. xanthus*) | SA7014 | this study | $\Delta pomX$; $\Delta pomZ$; $attB::P_{mxan0635}\ pomZ^{D90A}\text{-}mCh$ (pKA43) | Strain expressing PomZ$^{D90A}$-mCh in a *pomX* and *pomZ* deletion background |
| Strain, strain background (*M. xanthus*) | SA7061 | this study | $\Delta mglA$; $\Delta pomZ$; $\Delta pomX$; $attB::P_{mxan0635}\ mCh\text{-}pomX$ (pAH53) | Strain expressing mCh-PomX in a non-motile *pomX* and *pomZ* deletion background |
| Strain, strain background (*M. xanthus*) | SA7063 | this study | $\Delta pomZ$; $\Delta pomX$; $attB::P_{mxan0635}\ mCh\text{-}pomX$ (pAH53) | Strain expressing mCh-PomX in a *pomX* and *pomZ* deletion background |
| Strain, strain background (*M. xanthus*) | SA8240 | this study | $pomX::pomX^{K13AR15A}$; $\Delta pomZ$; $attB::P_{mxan0635}\ pomZ^{D90A}\text{-}mCh$ (pKA43) | Strain expressing PomZ$^{D90A}$-mCh in a *pomZ* deletion background with $pomX^{K13AR15A}$ mutation. |
| Strain, strain background (*M. xanthus*) | SA8250 | this study | $pomX::pomX^{K13AR15A}$; $\Delta pomY$; $attB::P_{pilA}\ pomY\text{-}mCh$ (pDS7) | Strain expressing PomZ$^{D90A}$-mCh in a $pomX^{K13AR15A}$ background |
| Strain, strain background (*M. xanthus*) | SA8268 | this study | $pomX::pomX^{K13AR15A}$; $\Delta pomY$; $\Delta pomZ$; $attB::P_{mxan0635}\ pomZ^{D90A}\text{-}mCh$ (pKA43) | Strain expressing PomZ$^{D90A}$-mCh in a *pomZ* and *pomY* deletion background with $pomX^{K13AR15A}$ mutation |
| Strain, strain background (*M. xanthus*) | SA9700 | this study | $pomX::pomX^{E6A}$ | Strain with a replacement of *pomX* with the $pomX^{E6A}$ allele |
| Strain, strain background (*M. xanthus*) | SA9701 | this study | $pomX::pomX^{Q7A}$ | Strain with a replacement of *pomX* with the $pomX^{Q7A}$ allele |
| Strain, strain background (*M. xanthus*) | SA9702 | this study | $pomX::pomX^{N8A}$ | Strain with a replacement of *pomX* with the $pomX^{N8A}$ allele |
| Strain, strain background (*M. xanthus*) | SA9714 | this study | $pomX::pomX^{R11A}$ | Strain with a replacement of *pomX* with the $pomX^{R11A}$ allele |
| Strain, strain background (*M. xanthus*) | SA9715 | this study | $pomX::pomX^{K3A}$ | Strain with a replacement of *pomX* with the $pomX^{K3A}$ allele |
| Strain, strain background (*M. xanthus*) | SA9716 | this study | $pomX::pomX^{R17A}$ | Strain with a replacement of *pomX* with the $pomX^{R17A}$ allele |
| Strain, strain background (*M. xanthus*) | SA9717 | this study | $pomX::pomX^{T22A}$ | Strain with a replacement of *pomX* with the $pomX^{T22A}$ allele |

*Continued on next page*

*Appendix 1—key resources table continued*

| Reagent type (species) or resource | Designation | Source or reference | Identifiers | Additional information |
|---|---|---|---|---|
| Strain, strain background (*M. xanthus*) | SA9718 | this study | *pomX::pomX*$^{K2A}$ | Strain with a replacement of *pomX* with the *pomX*$^{K2A}$ allele |
| Strain, strain background (*M. xanthus*) | SA9719 | this study | *pomX::pomX*$^{R15A}$ | Strain with a replacement of *pomX* with the *pomX*$^{R15A}$ allele |
| Strain, strain background (*M. xanthus*) | SA9720 | this study | Δ*pomY*; Δ*pomZ*; *attB*::P$_{mxan0635}$ *pomZ*$^{D90A}$-mCh (pKA43) | Strain expressing PomZ$^{D90A}$-mCh in a *pomZ* and *pomY* deletion background |
| Strain, strain background (*M. xanthus*) | SA9721 | this study | Δ*pomX*; Δ*pomY*; *attB*::P$_{pilA}$ *pomY-mCh* (pDS7) | Strain expressing PomY-mCh in a *pomY* and *pomX* deletion background |
| Strain, strain background (*M. xanthus*) | SA9726 | this study | Δ*pomX*; *attB*::P$_{mxan0635}$ *mCh-pomX*$^{N}$ (pDS252) | Strain expressing mCh-PomX$^{N}$ in a *pomX* deletion background |
| Strain, strain background (*M. xanthus*) | SA9727 | this study | Wild-type; *attB*::P$_{mxan0635}$ *mCh-pomX*$^{N}$ (pDS252) | Strain expressing mCh-PomX$^{N}$ in a WT background |
| Strain, strain background (*M. xanthus*) | SA9731 | this study | *pomX::pomX*$^{K13A}$ | Strain with a replacement of *pomX* with the *pomX*$^{K13A}$ allele |
| Strain, strain background (*M. xanthus*) | SA9732 | this study | *pomX::pomX*$^{S10A}$ | Strain with a replacement of *pomX* with the *pomX*$^{S10A}$ allele |
| Strain, strain background (*M. xanthus*) | SA9739 | this study | Δ*pomX*; *attB*::P$_{mxan0635}$ *mCh-pomX*$^{Q7A}$ (pDS317) | Strain expressing mCh-PomX$^{Q7A}$ in a *pomX* deletion background |
| Strain, strain background (*M. xanthus*) | SA9740 | this study | Δ*pomX*; *attB*::P$_{mxan0635}$ *mCh-pomX*$^{N8A}$ (pDS318) | Strain expressing mCh-PomX$^{N8A}$ in a *pomX* deletion background |
| Strain, strain background (*M. xanthus*) | SA9741 | this study | Δ*pomX*; *attB*::P$_{mxan0635}$ *mCh-pomX*$^{R17A}$ (pDS323) | Strain expressing mCh-PomX$^{R17A}$ in a *pomX* deletion background |
| Strain, strain background (*M. xanthus*) | SA9742 | this study | Δ*pomX*; *attB*::P$_{mxan0635}$ *mCh-pomX*$^{T22A}$ (pDS324) | Strain expressing mCh-PomX$^{T22A}$ in a *pomX* deletion background |
| Strain, strain background (*M. xanthus*) | SA9743 | this study | Δ*pomX*; *attB*::P$_{mxan0635}$ *mCh-pomX*$^{S10A}$ (pDS319) | Strain expressing mCh-PomX$^{S10A}$ in a *pomX* deletion background |
| Strain, strain background (*M. xanthus*) | SA9744 | this study | Δ*pomX*; *attB*::P$_{mxan0635}$ *mCh-pomX*$^{R11A}$ (pDS320) | Strain expressing mCh-PomX$^{R11A}$ in a *pomX* deletion background |
| Strain, strain background (*M. xanthus*) | SA9747 | this study | Δ*pomX*; *attB*::P$_{mxan0635}$ *mCh-pomX*$^{K13A}$ (pDS321) | Strain expressing mCh-PomX$^{K13A}$ in a *pomX* deletion background |
| Strain, strain background (*M. xanthus*) | SA9748 | this study | Δ*pomX*; *attB*::P$_{mxan0635}$ *mCh-pomX*$^{R15A}$ (pDS322) | Strain expressing mCh-PomX$^{R15A}$ in a *pomX* deletion background |
| Strain, strain background (*M. xanthus*) | SA9749 | this study | Δ*pomX*; *attB*::P$_{mxan0635}$ *mCh-pomX*$^{K2A}$ (pDS314) | Strain expressing mCh-PomX$^{K2A}$ in a *pomX* deletion background |
| Strain, strain background (*M. xanthus*) | SA9750 | this study | Δ*pomX*; *attB*::P$_{mxan0635}$ *mCh-pomX*$^{K3A}$ (pDS315) | Strain expressing mCh-PomX$^{K3A}$ in a *pomX* deletion background |

*Appendix 1—key resources table continued*

| Reagent type (species) or resource | Designation | Source or reference | Identifiers | Additional information |
|---|---|---|---|---|
| Strain, strain background (*M. xanthus*) | SA9751 | this study | Δ*pomX*; *attB*::P*mxan0635* mCh-pomX$^{E6A}$ (pDS316) | Strain expressing mCh-PomX$^{E6A}$ in a *pomX* deletion background |
| Strain, strain background (*M. xanthus*) | SA9752 | this study | Δ*pomX*; *attB*::P*mxan0635* mCh-pomX$^{K13AR15A}$ (pDS325) | Strain expressing mCh-PomX$^{K13AR15A}$ in a *pomX* deletion background |
| Strain, strain background (*M. xanthus*) | SA9753 | this study | Δ*mglA*; Δ*pomX*; *attB*:: P*mxan0635* mCh-pomX$^{K13AR15A}$ (pDS325) | Strain expressing mCh-PomX$^{K13AR15A}$ in a non-motile *pomX* deletion background |
| Strain, strain background (*M. xanthus*) | SA9754 | this study | Δ*mglA*; Δ*pomZ*; Δ*pomX*; P*mxan0635* mCh-pomX (pAH53); *mxan18-19*:: P*mxan0635* pomZ$^{D90A}$ (pDS80) | Strain expressing mCh-PomX in a non-motile *pomX* and *pomZ* deletion background that expresses PomZ$^{D90A}$. |
| Strain, strain background (*M. xanthus*) | SA9755 | this study | Wild-type; *attB*::P*mxan0635* mCh-pomX$^C$ (pDS329) | Strain expressing mCh-PomX$^C$ in a WT background |
| Strain, strain background (*M. xanthus*) | SA9756 | this study | Δ*pomY*; *attB*::P*mxan0635* mCh-pomX$^C$ (pDS329) | Strain expressing mCh-PomX$^C$ in a *pomY* deletion background |
| Strain, strain background (*M. xanthus*) | SA9757 | this study | Δ*pomZ*; *attB*::P*mxan0635* mCh-pomX$^C$ (pDS329) | Strain expressing mCh-PomX$^C$ in a *pomZ* deletion background |
| Strain, strain background (*M. xanthus*) | SA9762 | this study | Δ*pomX*; *attB*::P*mxan0635* mCh-pomX$^C$ (pDS329) | Strain expressing mCh-PomX$^C$ in a *pomX* deletion background |
| Recombinant DNA reagent | pAH27 (plasmid) | https://doi.org/10.1016/j.devcel.2017.04.011 | | Construct for in-frame deletion of *pomX*, KmR |
| Recombinant DNA reagent | pAH53 (plasmid) | https://doi.org/10.1016/j.devcel.2017.04.011 | | P*mxan0635* mCh-pomX, Mx8 *attB*, KmR |
| Recombinant DNA reagent | pDS1 (plasmid) | https://doi.org/10.1016/j.devcel.2017.04.011 | | Construct for in-frame deletion of *pomY*, KmR |
| Recombinant DNA reagent | pDS7 (plasmid) | https://doi.org/10.1016/j.devcel.2017.04.011 | | P*pilA* pomY-mCh, Mx8 *attB*, KmR |
| Recombinant DNA reagent | pDS16 (plasmid) | https://doi.org/10.1016/j.devcel.2017.04.011 | | Construct for in-frame deletion of *pomY* and *pomZ*, KmR |
| Recombinant DNA reagent | pDS80 (plasmid) | https://doi.org/10.1016/j.devcel.2017.04.011 | | P*mxan0635* pomZ$^{D90A}$, *mxan_18–19* intergenic region, TcR |
| Recombinant DNA reagent | pEMR3 (plasmid) | https://doi.org/10.1016/j.devcel.2017.04.011 | | Overexpression of PomX-His6, KmR |

*Continued on next page*

*Appendix 1—key resources table continued*

| Reagent type (species) or resource | Designation | Source or reference | Identifiers | Additional information |
|---|---|---|---|---|
| Recombinant DNA reagent | pKA1 (plasmid) | DOI: 10.1111/mmi.12094 | | Construct for in-frame deletion of pomZ, KmR |
| Recombinant DNA reagent | pKA3 (plasmid) | DOI: 10.1111/mmi.12094 | | Overexpression of His6-PomZ, KmR |
| Recombinant DNA reagent | pKA43 (plasmid) | DOI: 10.1111/mmi.12094 | | Pmxan0635 pomZD90A-mCh, Mx8 attB, TcR |
| Recombinant DNA reagent | pMAT12 (plasmid) | https://doi.org/10.1016/j.devcel.2017.04.011 | | Construct for in-frame deletion of pomZ and pomX, KmR |
| Recombinant DNA reagent | pAH152 (plasmid) | this study | | Overexpression of PomX$^C$-His$_6$, KmR |
| Recombinant DNA reagent | pSL16 (plasmid) | DOI: 10.1038/emboj.2011.291 | | Construct for in-frame deletion of mglA, KmR |
| Recombinant DNA reagent | pUT18 (plasmid) | https://doi.org/10.1073/pnas.95.10.5752 | | BACTH plasmid |
| Recombinant DNA reagent | pUT18C (plasmid) | https://doi.org/10.1073/pnas.95.10.5752 | | BACTH plasmid |
| Recombinant DNA reagent | pKT25 (plasmid) | https://doi.org/10.1073/pnas.95.10.5752 | | BACTH plasmid |
| Recombinant DNA reagent | pKNT25 (plasmid) | https://doi.org/10.1073/pnas.95.10.5752 | | BACTH plasmid |
| Recombinant DNA reagent | pAH154 (plasmid) | this study | | P$_{mxan0635}$ mCh-pomX$^N$, Mx8 attB, KmR |
| Recombinant DNA reagent | pAH157 (plasmid) | this study | | Overexpression of PomX$^N$-His$_6$, KmR |
| Recombinant DNA reagent | pAH165 (plasmid) | this study | | Overexpression of PomX$^{N\_K13AR15A}$-His$_6$, KmR |
| Recombinant DNA reagent | pDS100 (plasmid) | this study | | BACTH plasmid for pomZ (pUT18C), AmpR |
| Recombinant DNA reagent | pDS103 (plasmid) | this study | | BACTH plasmid for pomX (pUT18C), AmpR |
| Recombinant DNA reagent | pDS106 (plasmid) | this study | | BACTH plasmid for pomX (pKT25), KmR |
| Recombinant DNA reagent | pDS109 (plasmid) | this study | | BACTH plasmid for pomZ (pUT18), AmpR |
| Recombinant DNA reagent | pDS110 (plasmid) | this study | | BACTH plasmid for pomX (pUT18), AmpR |
| Recombinant DNA reagent | pDS114 (plasmid) | this study | | BACTH plasmid for pomX (pKNT25) KmR |

*Continued on next page*

*Appendix 1—key resources table continued*

| Reagent type (species) or resource | Designation | Source or reference | Identifiers | Additional information |
|---|---|---|---|---|
| Recombinant DNA reagent | pDS115 (plasmid) | this study | | BACTH plasmid for $pomZ^{D90A}$ (pUT18C), AmpR |
| Recombinant DNA reagent | pDS117 (plasmid) | this study | | BACTH plasmid for $pomZ^{D90A}$ (pUT18), AmpR |
| Recombinant DNA reagent | pDS120 (plasmid) | this study | | BACTH plasmid for $pomY$ (pUT18C), AmpR |
| Recombinant DNA reagent | pDS122 (plasmid) | this study | | BACTH plasmid for $pomY$ (pUT18), AmpR |
| Recombinant DNA reagent | pDS184 (plasmid) | this study | | BACTH plasmid for $pomX^{\Delta 2-21}$ (pUT18), AmpR |
| Recombinant DNA reagent | pDS185 (plasmid) | this study | | BACTH plasmid for $pomX^{\Delta 2-21}$ (pUT18C), AmpR |
| Recombinant DNA reagent | pDS186 (plasmid) | this study | | BACTH plasmid for $pomX^{\Delta 2-21}$ (pKT25), KmR |
| Recombinant DNA reagent | pDS187 (plasmid) | this study | | BACTH plasmid for $pomX^{\Delta 2-21}$ (pKNT25) KmR |
| Recombinant DNA reagent | pDS188 (plasmid) | this study | | BACTH plasmid for $pomX^{C}$ (pUT18), AmpR |
| Recombinant DNA reagent | pDS189 (plasmid) | this study | | BACTH plasmid for $pomX^{C}$ (pUT18C), AmpR |
| Recombinant DNA reagent | pDS190 (plasmid) | this study | | BACTH plasmid for $pomX^{C}$ (pKT25), KmR |
| Recombinant DNA reagent | pDS191 (plasmid) | this study | | BACTH plasmid for $pomX^{C}$ (pKNT25) KmR |
| Recombinant DNA reagent | pDS192 (plasmid) | this study | | BACTH plasmid for $pomX^{N}$ (pUT18), AmpR |
| Recombinant DNA reagent | pDS193 (plasmid) | this study | | BACTH plasmid for $pomX^{N}$ (pUT18C), AmpR |
| Recombinant DNA reagent | pDS194 (plasmid) | this study | | BACTH plasmid for $pomX^{N}$ (pKT25), KmR |
| Recombinant DNA reagent | pDS195 (plasmid) | this study | | BACTH plasmid for $pomX^{N}$ (pKNT25) KmR |
| Recombinant DNA reagent | pDS232 (plasmid) | this study | | Overexpression of PomX$^{N}$-Strep, KmR |
| Recombinant DNA reagent | pDS252 (plasmid) | this study | | P$_{mxan0635}$ $mCh$-$pomX^{N}$, Mx8 $attB$, KmR |
| Recombinant DNA reagent | pDS253 (plasmid) | this study | | BACTH plasmid for $pomX^{N\_K13AR15A}$ (pUT18), AmpR |
| Recombinant DNA reagent | pDS254 (plasmid) | this study | | BACTH plasmid for $pomX^{N\_K13AR15A}$ (pUT18C), AmpR |
| Recombinant DNA reagent | pDS255 (plasmid) | this study | | BACTH plasmid for $pomX^{N\_K13AR15A}$ (pKT25), KmR |
| Recombinant DNA reagent | pDS256 (plasmid) | this study | | BACTH plasmid for $pomX^{N\_K13AR15A}$ (pKNT25) KmR |
| Recombinant DNA reagent | pDS257 (plasmid) | this study | | BACTH plasmid for $pomX^{N\_\Delta 2-21}$ (pUT18), AmpR |
| Recombinant DNA reagent | pDS258 (plasmid) | this study | | BACTH plasmid for $pomX^{N\_\Delta 2-21}$ (pUT18C), AmpR |
| Recombinant DNA reagent | pDS259 (plasmid) | this study | | BACTH plasmid for $pomX^{N\_\Delta 2-21}$ (pKT25), KmR |

*Continued on next page*

*Appendix 1—key resources table continued*

| Reagent type (species) or resource | Designation | Source or reference | Identifiers | Additional information |
|---|---|---|---|---|
| Recombinant DNA reagent | pDS260 (plasmid) | this study | | BACTH plasmid for $pomX^{N\_\Delta2\text{-}21}$ (pKNT25) KmR |
| Recombinant DNA reagent | pDS303 (plasmid) | this study | | nat. site codon exchange for $pomX^{K2A}$, KmR |
| Recombinant DNA reagent | pDS304 (plasmid) | this study | | nat. site codon exchange for $pomX^{K3A}$, KmR |
| Recombinant DNA reagent | pDS305 (plasmid) | this study | | nat. site codon exchange for $pomX^{E6A}$, KmR |
| Recombinant DNA reagent | pDS306 (plasmid) | this study | | nat. site codon exchange for $pomX^{Q7A}$, KmR |
| Recombinant DNA reagent | pDS307 (plasmid) | this study | | nat. site codon exchange for $pomX^{N8A}$, KmR |
| Recombinant DNA reagent | pDS308 (plasmid) | this study | | nat. site codon exchange for $pomX^{S10A}$, KmR |
| Recombinant DNA reagent | pDS309 (plasmid) | this study | | nat. site codon exchange for $pomX^{R11A}$, KmR |
| Recombinant DNA reagent | pDS310 (plasmid) | this study | | nat. site codon exchange for $pomX^{K13A}$, KmR |
| Recombinant DNA reagent | pDS311 (plasmid) | this study | | nat. site codon exchange for $pomX^{R15A}$, KmR |
| Recombinant DNA reagent | pDS312 (plasmid) | this study | | nat. site codon exchange for $pomX^{R17A}$, KmR |
| Recombinant DNA reagent | pDS313 (plasmid) | this study | | nat. site codon exchange for $pomX^{T22A}$, KmR |
| Recombinant DNA reagent | pDS314 (plasmid) | this study | | $P_{mxan0635}$ $mCh\text{-}pomX^{K2A}$, Mx8 $attB$, KmR |
| Recombinant DNA reagent | pDS315 (plasmid) | this study | | $P_{mxan0635}$ $mCh\text{-}pomX^{K3A}$, Mx8 $attB$, KmR |
| Recombinant DNA reagent | pDS316 (plasmid) | this study | | $P_{mxan0635}$ $mCh\text{-}pomX^{E6A}$, Mx8 $attB$, KmR |
| Recombinant DNA reagent | pDS317 (plasmid) | this study | | $P_{mxan0635}$ $mCh\text{-}pomX^{Q7A}$, Mx8 $attB$, KmR |
| Recombinant DNA reagent | pDS318 (plasmid) | this study | | $P_{mxan0635}$ $mCh\text{-}pomX^{N8A}$, Mx8 $attB$, KmR |
| Recombinant DNA reagent | pDS319 (plasmid) | this study | | $P_{mxan0635}$ $mCh\text{-}pomX^{S10A}$, Mx8 $attB$, KmR |
| Recombinant DNA reagent | pDS320 (plasmid) | this study | | $P_{mxan0635}$ $mCh\text{-}pomX^{R11A}$, Mx8 $attB$, KmR |
| Recombinant DNA reagent | pDS321 (plasmid) | this study | | $P_{mxan0635}$ $mCh\text{-}pomX^{K13A}$, Mx8 $attB$, KmR |
| Recombinant DNA reagent | pDS322 (plasmid) | this study | | $P_{mxan0635}$ $mCh\text{-}pomX^{R15A}$, Mx8 $attB$, KmR |
| Recombinant DNA reagent | pDS323 (plasmid) | this study | | $P_{mxan0635}$ $mCh\text{-}pomX^{R17A}$, Mx8 $attB$, KmR |
| Recombinant DNA reagent | pDS324 (plasmid) | this study | | $P_{mxan0635}$ $mCh\text{-}pomX^{T22A}$, Mx8 $attB$, KmR |
| Recombinant DNA reagent | pDS325 (plasmid) | this study | | $P_{mxan0635}$ $mCh\text{-}pomX^{K13AR15A}$, Mx8 $attB$, KmR |
| Recombinant DNA reagent | pDS329 (plasmid) | this study | | $P_{mxan0635}$ $mCh\text{-}pomX^{C}$, Mx8 $attB$, KmR |

*Appendix 1—key resources table continued*

| Reagent type (species) or resource | Designation | Source or reference | Identifiers | Additional information |
|---|---|---|---|---|
| Recombinant DNA reagent | pDS333 (plasmid) | this study | | Overexpression of PomX$^C$-Strep, KmR |
| Recombinant DNA reagent | pEMR1 (plasmid) | this study | | Overexpression of PomY-His$_6$, KmR |
| Recombinant DNA reagent | pSH1 (plasmid) | this study | | nat. site codon exchange for pomX$^{K13AR15A}$, KmR |
| Recombinant DNA reagent | pSH36 (plasmid) | this study | | BACTH plasmid for pomX$^{K13AR15A}$ (pKNT25) KmR |
| Recombinant DNA reagent | pSH37 (plasmid) | this study | | BACTH plasmid for pomX$^{K13AR15A}$ (pKT25), KmR |
| Recombinant DNA reagent | pSH38 (plasmid) | this study | | BACTH plasmid for pomX$^{K13AR15A}$ (pUT18), AmpR |
| Recombinant DNA reagent | pSH39 (plasmid) | this study | | BACTH plasmid for pomX$^{K13AR15A}$ (pUT18C), AmpR |
| Recombinant DNA reagent | pSH58 (plasmid) | this study | | Overexpression of PomX$^{K13AR15A}$-His$_6$, KmR |
| Sequence-based reagent | pomX BTH fwd XbaI | this study | PCR primer | 5'-GCGTCTAGAGATGA AGAAAGCCTTTGAAC-3' |
| Sequence-based reagent | pomX BTH rev KpnI | this study | PCR primer | 5'-GCGGGTACCCGGC GCACCGTGGCCTGAC-3' |
| Sequence-based reagent | pomY BTH fwd XbaI | this study | PCR primer | 5'-GCGTCTAGAGGTGA GCGACGAGCGTCCG-3' |
| Sequence-based reagent | pomY BTH rev KpnI | this study | PCR primer | 5'-GCGGGTACCCGAG CGGCGAAGTATTTGTG-3' |
| Sequence-based reagent | pomZ BTH fwd XbaI | this study | PCR primer | 5'-GCGTCTAGAGATG GAAGCGCCGACGTAC-3' |
| Sequence-based reagent | pomZ BTH rev KpnI | this study | PCR primer | 5'-GCGGGTACCCGGCC GGCCTGCTGGGTGCC-3' |
| Sequence-based reagent | pomXΔ2–21 BTH fwd XbaI | this study | PCR primer | 5'-GCGTCTAGAGATGAC GGGCCTCGTCGACCCC-3' |
| Sequence-based reagent | pomXC BTH fwd XbaI | this study | PCR primer | 5'-GCGTCTAGAGATGGC CACCGTGGCGGAGGCG-3' |
| Sequence-based reagent | pomXN BTH rev KpnI | this study | PCR primer | 5'-GCGGGTACCCGGGGC AGCGGCTCCGGGCG-3' |
| Sequence-based reagent | 0636 up fwd | this study | PCR primer | 5'-GCGGGATCCGTC ACCCCAAGCCATTC-3' |
| Sequence-based reagent | PomX K2A rev native | this study | PCR primer | 5'-CAAAGGCTTT CGCCATGGTTCTCAG-3' |
| Sequence-based reagent | PomX K2A fwd native | this study | PCR primer | 5'-CTGAGAACCATGG CGAAAGCCTTTG-3' |

*Continued on next page*

*Appendix 1—key resources table continued*

| Reagent type (species) or resource | Designation | Source or reference | Identifiers | Additional information |
|---|---|---|---|---|
| Sequence-based reagent | 0636 HindIII rev stop | this study | PCR primer | 5'-GCGAAGCTTTCAGC GCACCGTGGCCTGAC-3' |
| Sequence-based reagent | PomX K3A rev native | this study | PCR primer | 5'-CTGTTCAAAGGC CGCCTTCATGGTTC-3' |
| Sequence-based reagent | PomX K3A fwd native | this study | PCR primer | 5'-GAACCATGAAGGCG GCCTTTGAACAG-3' |
| Sequence-based reagent | PomX E6A rev | this study | PCR primer | 5'-GGACACGTTCTG CGCAAAGGCTTTCTT-3' |
| Sequence-based reagent | PomX E6A fwd | this study | PCR primer | 5'-AAGAAAGCCTTTG CGCAGAACGTGTCC-3' |
| Sequence-based reagent | PomX Q7A rev | this study | PCR primer | 5'-GCGGGACACGTTCG CTTCAAAGGCTTT-3' |
| Sequence-based reagent | PomX Q7A fwd | this study | PCR primer | 5'-AAAGCCTTTGAAG CGAACGTGTCCCGC-3' |
| Sequence-based reagent | PomX N8A rev | this study | PCR primer | 5'-GGCGCGGGACACC GCCTGTTCAAAGGC-3' |
| Sequence-based reagent | PomX N8A fwd | this study | PCR primer | 5'-GCCTTTGAACA GGCGGTGTCCCGCGCC-3' |
| Sequence-based reagent | PomX S10A rev | this study | PCR primer | 5'-CGGCTTGGCGCGC GCCACGTTCTGTTC-3' |
| Sequence-based reagent | PomX S10A fwd | this study | PCR primer | 5'-GAACAGAACGTGG CGCGCGCCAAGCCG-3' |
| Sequence-based reagent | PomX R11A rev | this study | PCR primer | 5'-GCGCGGCTTGG CCGCGGACACGTTCTG-3' |
| Sequence-based reagent | PomX R11A fwd | this study | PCR primer | 5'-CAGAACGTGTCC GCGGCCAAGCCGCGC-3' |
| Sequence-based reagent | PomX K13A rev | this study | PCR primer | 5'-GCGGAGGCGCGG CGCGGCGCGGGACAC-3' |
| Sequence-based reagent | PomX K13A fwd | this study | PCR primer | 5'-GTGTCCCGCGCCGCG CCGCGCCTCCGC-3' |
| Sequence-based reagent | PomX R15A rev | this study | PCR primer | 5'-GCCCAGGCGGAGCGC CGGCTTGGCGCG-3' |
| Sequence-based reagent | PomX R15A fwd | this study | PCR primer | 5'-CGCGCCAAGCCGGC GCTCCGCCTGGGC-3' |
| Sequence-based reagent | PomX R17A rev | this study | PCR primer | 5'-CAGCGCGCCCAGCG CGAGGCGCGGCTT-3' |

*Continued on next page*

*Appendix 1—key resources table continued*

| Reagent type (species) or resource | Designation | Source or reference | Identifiers | Additional information |
|---|---|---|---|---|
| Sequence-based reagent | PomX R17A fwd | this study | PCR primer | 5'-AAGCCGCGCCTCG CGCTGGGCGCGCTG-3' |
| Sequence-based reagent | PomX T22A rev | this study | PCR primer | 5'-GTCGACGAGGC CCGCCAGCGCGCCCAG-3' |
| Sequence-based reagent | PomX T22A fwd | this study | PCR primer | 5'-CTGGGCGCGCT GGCGGGCCTCGTCGAC-3' |
| Sequence-based reagent | PomX K13AR15A rev | this study | PCR primer | 5'-CAGAACGTGTCCCGCG CCGCGCCGGCCCTCCG CCTGGGCGCGCTG-3' |
| Sequence-based reagent | PomX K13AR15A fwd | this study | PCR primer | 5'-CAGCGCGCCCAGGCG GAGGGCCGGCGCGGCG CGGGACACCTTCTG-3' |
| Sequence-based reagent | mCherry XbaI fwd | this study | PCR primer | 5'-GCGTCTAGAGT GAGCAAGGGCGAGGAG-3' |
| Sequence-based reagent | PomX K2A rev | this study | PCR primer | 5'-TTCAAAGGCTTTC GCCATGGCTCCGCC-3' |
| Sequence-based reagent | PomX K2A fwd | this study | PCR primer | 5'-GGCGGAGCCAT GGCGAAAGCCTTTGAA-3' |
| Sequence-based reagent | KA348 | this study | PCR primer | 5'-GCCAAGCTTTC AGCGCACCGTGGCCTG-3' |
| Sequence-based reagent | PomX K3A fwd | this study | PCR primer | 5'-GGAGCCATGAAG GCGGCCTTTGAACAG-3' |
| Sequence-based reagent | PomX K3A rev | this study | PCR primer | 5'-CTGTTCAAAGGCC GCCTTCATGGCTCC-3' |
| Sequence-based reagent | AH142 | this study | PCR primer | 5'-GGAATTCCATATG GCCACCGTGGCGGAGGCG-3' |
| Sequence-based reagent | KA346 | this study | PCR primer | 5'-GCCAAGCTTGCGCA CCGTGGCCTGACTC-3' |
| Sequence-based reagent | AH143 | this study | PCR primer | 5'-CCCAAGCTTGG GCAGCGGCTCCGGGCG-3' |
| Sequence-based reagent | NdeI PomX fwd | this study | PCR primer | 5'-GGAATTCCATATGA AGAAAGCCTTTGAACAG-3' |
| Sequence-based reagent | AH144 | this study | PCR primer | 5'-GCCAAGCTTTCAG GGCAGCGGCTCCGGGCG-3' |
| Sequence-based reagent | KA384 | this study | PCR primer | 5'-GCGGGATCCGGC GGAGCCATGAAGAA AGCCTTTGAACAG-3' |
| Sequence-based reagent | DS276 | this study | PCR primer | 5'-GCGAAGCTTACTTC TCGAACTGTGGGTGACTC CAGCGCACCGTGGCCTGAC-3' |

*Continued on next page*

*Appendix 1—key resources table continued*

| Reagent type (species) or resource | Designation | Source or reference | Identifiers | Additional information |
|---|---|---|---|---|
| Sequence-based reagent | DS277 | this study | PCR primer | 5'-GCGCCATGGCC ACCGTGGCGGAGGCG-3' |
| Sequence-based reagent | PomX BspHI fwd | this study | PCR primer | 5'-GCGTCATGAAGAA AGCCTTTGAACAGAACG-3' |
| Sequence-based reagent | PomXN rev strep-tag | this study | PCR primer | 5'-GCGAAGCTTACTTCTC GAACTGTGGGTGACTCC AGGGCAGCGGCTCCGGGCG-3' |
| Sequence-based reagent | NdeI-PomY fwd | this study | PCR primer | 5'-GGAATTCCATATGA GCGACGAGCGTCCGGAC-3' |
| Sequence-based reagent | PomY C-term his rev | this study | PCR primer | 5'-CGGAAGCTTAGCG GCGAAGTATTTGTGC-3' |
| Sequence-based reagent | AH141 | this study | PCR primer | 5'-GCGGGATCCGGCGG AGCCGCCACCGTGGCGGAGGCG-3' |
| Antibody | α-PomX (rabbit, polyclonal) | https://doi.org/10.1016/j.devcel.2017.04.011 | | Western Blot (1:15000) |
| Antibody | α-PomY (rabbit, polyclonal) | https://doi.org/10.1016/j.devcel.2017.04.011 | | Western Blot (1:15000) |
| Antibody | α-PomZ (rabbit, polyclonal) | DOI: 10.1111/mmi.12094 | | Western Blot (1:10000) |
| Antibody | α-PilC (rabbit, polyclonal) | DOI:10.1111/j.1365–2958.2009.06891.x | | Western Blot (1:3000) |
| Antibody | α-mCherry (rabbit, polyclonal) | Biovision | Cat# 5993 | Western Blot (1:10000) |
| Antibody | horseradish-conjugated α-rabbit immunoglobulin G (goat,polyclonal) | Sigma-Aldrich | Cat# A0545-1ML | Western Blot (1:25000) |
| Peptide, recombinant protein | PomX$^{NPEP}$ | Thermo Scientific | | MKKAFEQNVSRAKPRLRLGALT |
| Peptide, recombinant protein | PomX$^{NPEPK13AR15A}$ | Thermo Scientific | | MKKAFEQNVSRAAPALRLGALT |
| Software, algorithm | Metamorph_v 7.5 | Molecular Devices | | |
| Software, algorithm | Oufti | DOI: 10.1111/mmi.13264 | | http://www.oufti.org/ |
| Software, algorithm | Matlab R2018a | MathWorks | | |
| Commercial assay or kit | Luminata Forte | Fisher scientific | Cat# 10394675 | |

