## [Decision Letter]

**Acceptance summary:**

This study examines a bacterial system that positions the cell division septum in *Myxococcus xanthus*, called PomXYZ, a tripartite protein-protein interaction system. It will be of broad interest to readers who are interested in how macromolecular complexes and DNA are dynamically positioned inside cells.

**Decision letter after peer review:**

Thank you for submitting your article "PomX, a ParA/MinD ATPase activating protein, is a triple regulator of cell division in *Myxococcus xanthus*" for consideration by *eLife*. Your article has been reviewed by three peer reviewers, and the evaluation has been overseen by Gisela Storz serving as the Reviewing and Senior Editor. The following individual involved in review of your submission has agreed to reveal their identity: Joe Lutkenhaus (Reviewer #3).

Essential Revisions:

The reviewers, who were all enthusiastic about this study, have several comments for improving this manuscript.

1) Given that PomX polymerizes so readily, what is the form of this protein in the assays? Polymerized or not and how does that affect the results.

2) Related to the first point, how is PomX polymerization is regulated? Is it just by buffer conditions?

3) In the text it is stated that PomY also stimulates the ATPase activity of PomZ. Is that really the case? Does it have an N-terminal are like PomX?

4) The K3A mutation (which behaves as a null mutation) lowers the intracellular level of PomX below that detectable, which is understandable. However this ignores a potentially critical residue. The role of the residue could be tested using peptides (as was done with K13AR15A). ATPase assays with a K3A peptide would be useful information and might uncover a critical residue.

5) Some of the strong conclusions about ATP hydrolysis stimulation need to be tempered.

Reviewer #1 (Recommendations for the authors):

I have some suggestions that should be straightforward for the authors to incorporate if they wish.

1) Subsection “PomX AAP activity resides in PomX^N^”. The authors comment on the cooperativity of the ATPase activity of PomZ and make a statement on the speed of the reaction relative to other ATPases. It appears that the reactions in Figure 2E-H were not performed in a condition where one substrate was limiting (thus, the reactions were unable to reach a plateau, as stated in the aforementioned subsection). As such, the reaction did not reach saturation, so it is difficult to draw conclusions regarding the shape of the curve (i.e., first order, second order, etc) and therefore whether the hydrolysis is cooperative. Since the reaction did not produce a classical Michaelis-Menten curve, it is also not possible to calculate a kcat value (turnover) for the reaction (mol ATP hydrolyzed/min/mol protein), which would allow a conclusion about how “slow” the enzyme is. One way to achieve this is to hold the protein concentration constant (low) and vary the nucleotide level. In lieu of repeating these experiments, since these conclusions are not central to the authors' overall story, the authors may consider omitting the specific conclusions regarding enzyme kinetics.

Reviewer #2 (Recommendations for the authors):

1) The authors show that PomX, as in other AAPs, stimulates ATP hydrolysis via a short N-terminal segment that can be replaced by peptides in vitro. They conclude that K13 and R15 are important residues for the PomZ interaction, but admit that they cannot be the only ones since the K13AR15A mutant still retains partial function. But they do not discuss K3 because the K3A mutation (which behaves as a null mutation) lowers the intracellular level of PomX below that detectable, which is understandable. However this ignores a potentially critical residue and they could test the role of the residue using peptides (as they did with K13AR15A). ATPase assays with a K3A peptide would be useful information and might uncover a critical residue.

2) The ParA ATPase has also been implicated in splitting ParB complexes/foci (see Ah-Seng et al., 2013), which extends similarities with ParA-like systems and could be mentioned.

3) I found Figure 3D, right panel, confusing and found it difficult to discern what data points were plotted and what they mean. For example, what does the % mean where labelled directly on the graph? More explanation in the legend could be included, or the data plotted in a more simplified representation.

Reviewer #3 (Recommendations for the authors):

The authors hone in on the N-ter peptide of PomX and find that it is necessary for interaction with PomZ as well as ability to stimulate the ATPase activity of PomZ.

Abstract: I might reword the fifth sentence. The C-terminal domain interacts with PomY and forms polymers which serve as a scaffold for PomXYZ complex formation.

The focus in this paper is on PomX but it is not clear why since PomY has also been shown to stimulate the ATPase activity of PomZ to a similar degree (Introduction). Some rationale should be provided for focusing on PomX. Is the stimulation by PomY reproducible? Does it have a role? Seems incomplete without this.

Introduction – triple regulator – why not say PomX displays three activities required to regulate cell division – AAP, scaffold and along with PomZ is required for fission of the complex.

Figure 1—figure supplement 1B. What are the extra bands in the rightmost Western panel when a-mCh antibody is used? The bands at 72kDa and ~45kDa do not show up (or are less intense) in the left and center panels with the same antibody.

Results. ∆pomX cells have only 2X the average length but produce minicells. Does that mean the frequency of cell division is not affected? Or are not many minicells produced?

Results. From Figure 2A it appears that the BATCH for the interaction between PomZ and PomX^N^ only works well in one orientation. Does T18 attached to the N-ter of PomX interfere with the interaction? T25 does not seem to interfere. A comment about this should appear somewhere.

Results. Spontaneous filament formation by PomX and PomX^C^. How is assembly regulated in vitro. What is added to the solution to induce assembly? Is it induced by changing the salt concentration? When PomX or PomX^C^ are isolated are filaments already present? Perhaps this has been addressed in an earlier publication but I did not see it.

In the ATPase assays what state is PomX in? Is it polymerized? Does it make a difference whether it is or not.

Figure 5A. Any reason why PomX^N^ was not included as a control?

"all four PomX^NPEP^ mutants". Aren't there only two mutants? Each containing two changes.

Is the NPEP predicted to be an α helix. The N-terminus of MinE bound to MInD is an α helix so I would expect this to be the same although it does not have to be.

Discussion. It seems in an earlier publication there was stimulation of the ATPase activity by both PomX and PomY in the absence of DNA. Please clarify.

Subsection “The PomX/PomZ interaction is important for PomX/Y/Z cluster fission during division”. What is the consequence of not splitting the cluster?

Discussion. Although the precise mechanism is unknown, a mechanism has been proposed for MinD-MInE as well as a comparison with RAS in Park et al., 2012.

Can the authors comment on parallels to ParA/ParB where ParB is bound to DNA to presumably have a large cluster of ParB (the AAP) whereas in the system studied here PomX polymerizes to form large clusters. This is presumably needed to migrate on the DNA.

---

## [Author Response]

Essential Revisions:The reviewers, who were all enthusiastic about this study, have several comments for improving this manuscript.1) Given that PomX polymerizes so readily, what is the form of this protein in the assays? Polymerized or not and how does that affect the results.

Thank you for drawing our attention to these questions! Purified PomX-His_6,_ as mentioned in the comment below, forms filaments/polymerizes under all conditions tested in vitro. The buffer conditions for the ATPase assay, sedimentation assays, and the EM analyses are identical. Moreover, the concentration range of PomX used in these different assays is the same. This also means that the protein variants of PomX and PomX^C^ that are added to the ATPase assays are polymers. We would like to add that in all the assays, we very carefully monitor the amount of PomX added to be able to make clear statements about the concentration of PomX. Importantly, PomX^N^, which does not form polymers/filaments, stimulates PomZ ATPase activity as efficiently as PomX full-length protein. These observations support that the PomX^N^ domains in a PomX polymer act independently of each other to stimulate PomZ ATPase activity. In the revised manuscript, we have modified the text to emphasize this. We return to this point in the Discussion.

2) Related to the first point, how is PomX polymerization is regulated? Is it just by buffer conditions?

Upon (over)production of PomX in *E. coli*, it spontaneously forms filaments under all conditions tested in vitro. We have used different tags, attached the tag to the N-terminus or C-terminus, tried different expression levels, used different temperatures during expression, and used different buffer conditions for purifying the protein. Under all these different conditions, PomX always purifies as a polymer. Even upon expression of native PomX in *E. coli* it purifies as a polymer. Also, upon expression of mCh-tagged PomX in *M. xanthus*, the protein forms a filamentous structure. Therefore, we have not been able to identify any physiological condition where PomX does not form a polymer/filament. In other words, polymerization is spontaneous and independent of any cofactor (that we know of). Therefore, we have as a working model that PomX upon synthesis spontaneously polymerizes and that this polymerization is not regulated. The buffer conditions for the ATPase assay, sedimentation assays, and the EM analyses are identical. Moreover, the concentration range of PomX used in these different assays is the same. This also means that the protein variants of PomX and PomX^C^ that are added to the ATPase assays are polymers. We would like to add that in all the assays, we very carefully monitor the amount of PomX added to be able to make clear statements about the concentration of PomX. To make clear that PomX spontaneously polymerizes in a cofactor-independent manner, we have modified the text.

3) In the text it is stated that PomY also stimulates the ATPase activity of PomZ. Is that really the case? Does it have an N-terminal are like PomX?

In the revised manuscript, we have now included a brief description of the rationale for focusing our analysis on PomX in this manuscript. To understand the mechanism of the PomX/Y/Z system we are of course also working on PomY (which we can assure and confirm has AAP activity as previously published); but that work has not been finished yet. We have also added that “Interestingly, PomY, the second AAP of PomZ (Schumacher et al., 2017), does not have an N-terminus enriched in positively charged residues suggesting that its mode of action could be different from that of previously described AAPs”.

4) The K3A mutation (which behaves as a null mutation) lowers the intracellular level of PomX below that detectable, which is understandable. However this ignores a potentially critical residue. The role of the residue could be tested using peptides (as was done with K13AR15A). ATPase assays with a K3A peptide would be useful information and might uncover a critical residue.

We apologize for this confusion. The untagged PomX^K3A^ variant does not accumulate in *M. xanthus* (and therefore does not complement the Δ*pomX* mutant). However, the mCh-tagged PomX^K3A^ variant accumulates in *M. xanthus* at a level similar to that of the WT protein and complements the cell length and division defects of the Δ*pomX* mutant (Shown in Figure 4—figure supplement 1A, B). Therefore, we conclude that the Lys3 residue is not important for PomX function. To make this conclusion more clear, we have modified the text. Moreover, we included the cell length of the Δ*pomX* mutant expressing mCh-PomX^K3A^. Because the mCh-PomX^K3A^ variant is fully active, we have not done any experiments to specifically explore the function of Lys3 in the activation of PomZ ATPase activity.

5) Some of the strong conclusions about ATP hydrolysis stimulation need to be tempered.

Thanks for pointing this out to us. Throughout the manuscript, we have deleted comments and conclusions about enzyme kinetics.

Reviewer #1 (Recommendations for the authors):I have some suggestions that should be straightforward for the authors to incorporate if they wish.1) Subsection “PomX AAP activity resides in PomX^N^”. The authors comment on the cooperativity of the ATPase activity of PomZ and make a statement on the speed of the reaction relative to other ATPases. It appears that the reactions in Figure 2E-H were not performed in a condition where one substrate was limiting (thus, the reactions were unable to reach a plateau, as stated in the aforementioned subsection). As such, the reaction did not reach saturation, so it is difficult to draw conclusions regarding the shape of the curve (i.e., first order, second order, etc) and therefore whether the hydrolysis is cooperative. Since the reaction did not produce a classical Michaelis-Menten curve, it is also not possible to calculate a kcat value (turnover) for the reaction (mol ATP hydrolyzed/min/mol protein), which would allow a conclusion about how “slow” the enzyme is. One way to achieve this is to hold the protein concentration constant (low) and vary the nucleotide level. In lieu of repeating these experiments, since these conclusions are not central to the authors' overall story, the authors may consider omitting the specific conclusions regarding enzyme kinetics.

Thanks for pointing this out to us. Throughout the manuscript we have deleted comments and conclusions about enzyme kinetics.

Reviewer #2 (Recommendations for the authors):1) The authors show that PomX, as in other AAPs, stimulates ATP hydrolysis via a short N-terminal segment that can be replaced by peptides in vitro. They conclude that K13 and R15 are important residues for the PomZ interaction, but admit that they cannot be the only ones since the K13AR15A mutant still retains partial function. But they do not discuss K3 because the K3A mutation (which behaves as a null mutation) lowers the intracellular level of PomX below that detectable, which is understandable. However this ignores a potentially critical residue and they could test the role of the residue using peptides (as they did with K13AR15A). ATPase assays with a K3A peptide would be useful information and might uncover a critical residue.

We apologize for this confusion. The untagged PomX^K3A^ variant does not accumulate in *M. xanthus* (and therefore does not complement the Δ*pomX* mutant). However, the mCh-tagged PomX^K3A^ variant accumulates in *M. xanthus* at a level similar to that of the WT protein and complements the cell length and division defects of the Δ*pomX* mutant (Shown in Figure 4—figure supplement 1A, B). Therefore, we conclude that the Lys3 residue is not important for PomX function. To make this conclusion more clear, we have modified the text. Moreover, we included the cell length of the Δ*pomX* mutant expressing mCh-PomX^K3A^. Because the mCh-PomX^K3A^ variant is fully active, we have not done any experiments to specifically explore the function of Lys3 in the activation of PomZ ATPase activity.

2) The ParA ATPase has also been implicated in splitting ParB complexes/foci (see Ah-Seng et al., 2013), which extends similarities with ParA-like systems and could be mentioned.

Thank you for pointing this out to us. We have now included a brief comparison to the SopA/B system in the Discussion.

3) I found Figure 3D, right panel, confusing and found it difficult to discern what data points were plotted and what they mean. For example, what does the % mean where labelled directly on the graph? More explanation in the legend could be included, or the data plotted in a more simplified representation.

We apologize for this confusion. We use Figure 3D right panel to compare cell length, position of cell division constrictions and the constriction frequency. To make the figure more accessible, we now precisely refer to the left and right panels of this figure in the main text. In addition, we changed the figure legend to explain what we are plotting more carefully.

Reviewer #3 (Recommendations for the authors):The authors hone in on the N-ter peptide of PomX and find that it is necessary for interaction with PomZ as well as ability to stimulate the ATPase activity of PomZ.Abstract: I might reword the fifth sentence. The C-terminal domain interacts with PomY and forms polymers which serve as a scaffold for PomXYZ complex formation.

Thanks and changed as suggested.

The focus in this paper is on PomX but it is not clear why since PomY has also been shown to stimulate the ATPase activity of PomZ to a similar degree (Introduction). Some rationale should be provided for focusing on PomX. Is the stimulation by PomY reproducible? Does it have a role? Seems incomplete without this.

In the revised manuscript, we have now included a brief description of the rationale for focusing our analysis on PomX in this manuscript. To understand the mechanism of the PomX/Y/Z system we are of course also working on PomY (which we can assure and confirm has AAP activity as previously published); but that work has not been finished yet. We also think that the manuscript would be far too long to include data on how PomY activates PomZ ATPase activity. We have added that “Interestingly, PomY, the second AAP of PomZ (Schumacher et al., 2017), does not have an N-terminus enriched in positively charged residues suggesting that its mode of action could be different from that of previously described AAPs”.

Introduction – triple regulator – why not say PomX displays three activities required to regulate cell division – AAP, scaffold and along with PomZ is required for fission of the complex.

Changed essentially as suggested (Introduction).

Figure 1—figure supplement 1B. What are the extra bands in the rightmost Western panel when a-mCh antibody is used? The bands at 72kDa and ~45kDa do not show up (or are less intense) in the left and center panels with the same antibody.

The bands at approximately 72kDa and 45kDa are unspecific bands that sometimes and sometimes not appear in Western blots with α-mCherry antibodies. Please note that these bands are also present in the cell extract of the WT and Δ*pomX* strains, none of which express an mCherry protein. We are struggling to find out why these bands sometimes appear. We would like to add that the presence of these bands does not affect any of our conclusions. In the revised manuscript, we have clearly marked these bands in Figure 1—figure supplement 1B and explained in the legend that “Note that the three bands labeled * in the right and left α-mCh Western blot of B are unspecific bands that sometimes appear in the Western blots with α-mCh antibodies”.

Results. ∆pomX cells have only 2X the average length but produce minicells. Does that mean the frequency of cell division is not affected? Or are not many minicells produced?

The constriction frequency is reduced approximately 5-fold in the Δ*pomX* mutant compared to the WT. Moreover, the constrictions are not only at midcell. Altogether, this results in the formation of filamentous cells and minicells. Importantly, the variance in cell length of the filamentous cells in the Δ*pomX* mutant (see Figure 1D) is much large compared to WT and some cells have a length >40 µm as would be expected with a 5-fold reduction of the constriction frequency.

Results. From Figure 2A it appears that the BATCH for the interaction between PomZ and PomX^N^ only works well in one orientation. Does T18 attached to the N-ter of PomX interfere with the interaction? T25 does not seem to interfere. A comment about this should appear somewhere.

As shown in Figure 2A, the interaction between PomZ and PomX depends on where the tag (T18) is added to PomZ but is independent of where the tag (T25) is added to PomX and PomX^N^. We have not tested whether T18 attached to the N-ter of PomX interferes with the interaction to PomZ; but we would expect that this is not the case because T25 added to the N-terminus of PomX does not interfere with its interaction with PomZ. It is quite common that interactions observed in BACTH assays depend on where the tag is added. So, the important take-home message from the BACTH is that PomX and PomX^N^ interact with PomZ. Therefore, we decided – and we hope that you agree with this decision – not to include a comment about the interactions that we did not see. Instead, we went on to check for interactions in vitro using purified proteins.

Results. Spontaneous filament formation by PomX and PomX^C^. How is assembly regulated in vitro. What is added to the solution to induce assembly? Is it induced by changing the salt concentration? When PomX or PomX^C^ are isolated are filaments already present? Perhaps this has been addressed in an earlier publication but I did not see it.

Upon (over)production of PomX in *E. coli*, it spontaneously forms filaments under all conditions tested in vitro. We have used different tags, attached the tag to the N-terminus or C-terminus, tried different expression levels, used different temperatures during expression, and used different buffer conditions for purifying the protein. Under all these different conditions, PomX always purifies as a polymer. Even upon expression of native PomX in *E. coli* it purifies as a polymer. Also, upon expression of mCh-tagged PomX in *M. xanthus*, the protein forms a filamentous structure. Therefore, we have not been able to identify any physiological condition where PomX does not form a polymer/filament. In other words, polymerization is spontaneous and independent of any cofactor (that we know of). Therefore, we have as a working model that PomX upon synthesis spontaneously polymerizes and that this polymerization is not regulated. The buffer conditions for the ATPase assay, sedimentation assays, and the EM analyses are identical. Moreover, the concentration range of PomX used in these different assays is the same. This also means that the protein variants of PomX and PomX^C^ that are added to the ATPase assays are polymers. We would like to add that in all the assays we very carefully monitor the amount of PomX added to be able to make clear statements about the concentration of PomX. To make clear that PomX spontaneously polymerizes in a cofactor-independent manner, we have modified the text.

In the ATPase assays what state is PomX in? Is it polymerized? Does it make a difference whether it is or not.

Thank you for drawing our attention to these questions! Purified PomX-His_6,_ as mentioned in the comment above, forms filaments/polymerizes under all conditions tested in vitro. The buffer conditions for the ATPase assay, sedimentation assays, and the EM analyses are identical. Moreover, the concentration range of PomX used in these different assays is the same. This also means that the protein variants of PomX and PomX^C^ that are added to the ATPase assays are polymers. We would like to add that in all the assays, we very carefully monitor the amount of PomX added to be able to make clear statements about the concentration of PomX. Importantly, PomX^N^, which does not form polymers/filaments, stimulates PomZ ATPase activity as efficiently as PomX full-length protein. These observations support that the PomX^N^ domains in a PomX polymer act independently of each other to stimulate PomZ ATPase activity. In the revised manuscript, we have modified the text to emphasize this. We return to this point in the Discussion.

Figure 5A. Any reason why PomX^N^ was not included as a control?

We decided not to include PomX^N^ as a control because (1) it is already included in Figure 2A, and (2) to keep the figure simple. We have changed the legend to Figure 5A to make clear that all these experiments were done in parallel and can be compared directly.

"all four PomX^NPEP^ mutants". Aren't there only two mutants? Each containing two changes.

We apologize for the confusion. It actually is four variants and we have clarified that in the text.

Is the NPEP predicted to be an α helix. The N-terminus of MinE bound to MInD is an α helix so I would expect this to be the same although it does not have to be.

Predictions programs (Psipred, Predictprotein and JPred4) that identify an α-helix in the N-terminus of MinE do not predict an α-helix in PomX^NPEP^.

Discussion. It seems in an earlier publication there was stimulation of the ATPase activity by both PomX and PomY in the absence of DNA. Please clarify.

Using PomX-His_6_ and PomY-His_6_ and a MalE-PomZ protein in a malachite-green colorimetric endpoint measurement, we previously reported that PomX-His_6_ and PomY-His_6_ weakly but significantly stimulate PomZ ATPase activity in the absence of DNA (Schumacher et al., 2017). Moreover, we reported that PomX-His_6_ and PomY-His_6_ stimulated PomZ ATPase activity strongly in the presence of DNA. Thus, overall our previous results and those reported in the current manuscript are in agreement. In the current manuscript, we purified PomZ-His_6_ (instead of MalE-PomZ) to improve and increase protein yield and purity. Along the same lines, we used an NADH-coupled enzymatic assay that allows us to measure ADP release in real-time and which is not susceptible to perturbation by free phosphate in the buffers used. We believe that these changes explain the slight difference between our previous results and the ones that we report here.

Subsection “The PomX/PomZ interaction is important for PomX/Y/Z cluster fission during division”. What is the consequence of not splitting the cluster?

Thanks for pointing out that we had not addressed this in the original version of the manuscript. We have now included that “We speculate that the reduced frequency of cluster fission events contributes to the cell division defect in the PomX AAP mutants”.

Discussion. Although the precise mechanism is unknown, a mechanism has been proposed for MinD-MInE as well as a comparison with RAS in Park et al., 2012.

Thanks for pointing this out to us. We have included mentioning of this proposed mechanism in the Discussion.

Can the authors comment on parallels to ParA/ParB where ParB is bound to DNA to presumably have a large cluster of ParB (the AAP) whereas in the system studied here PomX polymerizes to form large clusters. This is presumably needed to migrate on the DNA.

Thanks for this suggestion. We have now included a comparison between the PomX/Y complex, ParB-*parS* complexes, cytoplasmic chemoreceptor clusters and carboxysomes to point out that they may not only be similar in terms of the mechanism of the AAP but also in terms of translocation on the nucleoid (Discussion).